# Multimodal Remote Sensing Image Registration Methods and Advancements: A Survey

**Xinyue Zhang** [1]**, Chengcai Leng** [1,*]**, Yameng Hong** [1]**, Zhao Pei** [2]**, Irene Cheng** [3]** and Anup Basu** [3]

1   School of Mathematics, Northwest University, Xi'an 710127, China; zhangxinyue@stumail.nwu.edu.cn (X.Z.); hongyameng@stumail.nwu.edu.cn (Y.H.)
2   School of Computer Science, Shaanxi Normal University, Xi'an 710119, China; zpei@snnu.edu.cn
3   Department of Computing Science, University of Alberta, Edmonton, AB T6G 2E8, Canada; locheng@ualberta.ca (I.C.); basu@ualberta.ca (A.B.)
*   Correspondence: ccleng@nwu.edu.cn

**Abstract:** With rapid advancements in remote sensing image registration algorithms, comprehensive imaging applications are no longer limited to single-modal remote sensing images. Instead, multi-modal remote sensing (MMRS) image registration has become a research focus in recent years. However, considering multi-source, multi-temporal, and multi-spectrum input introduces significant nonlinear radiation differences in MMRS images for which researchers need to develop novel solutions. At present, comprehensive reviews and analyses of MMRS image registration methods are inadequate in related fields. Thus, this paper introduces three theoretical frameworks: namely, area-based, feature-based and deep learning-based methods. We present a brief review of traditional methods and focus on more advanced methods for MMRS image registration proposed in recent years. Our review or comprehensive analysis is intended to provide researchers in related fields with advanced understanding to achieve further breakthroughs and innovations.

**Keywords:** MMRS image registration; area-based methods; feature-based methods; deep-learning based methods

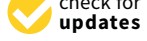



## 1. Introduction

Image registration is the process of geometrical alignment and matching of two or more images of the same scene, acquired from different sensors, with different views and at different times [1]. In the field of remote sensing, the accuracy of image registration plays a crucial role in subsequent applications, such as image fusion, map correction and change analysis [2,3]. It is possible to obtain multi-source remote sensing image data with the rapid development of aerospace technology and remote sensing. Comprehensive utilization of multi-source remote sensing data has been widely used to realize the uniqueness and complementarity of different remote sensing images, in order to acquire images containing more information.

The registration of MMRS images still has difficulties in applications due to significant geometric distortion and nonlinear intensity differences between these images. However, it is necessary to integrate these images for earth observation applications [4]. For instance, different sensors capture optical and synthetic aperture radar (SAR) images, and different imaging mechanisms produce distinct characteristics of an area. Therefore, fusing two types of images is more conducive to representing a given area. Simultaneously, when encountering emergencies, such as weather disasters, only SAR images are useful since they can work during both day and night and see-through cloud and fog to capture images. In this case, there is an inevitable problem in combining traditional optical images with a currently acquired SAR image to analyze the imaged area [5].

Although significant advances have been made in automatic image registration technologies in the past few decades, due to the high-performance requirements of MMRS

image registration, scholars strive to propose technologies advancing the state-of-the-art to address the problems of geometric differences and time efficiency.

The field of remote sensing can be roughly divided into remote sensing image acquisition technology and remote sensing information processing technology. Photogrammetry technology that mainly relies on aerospace and ground imaging platforms is different from the remote sensing technology that relies on satellite platforms. There are also differences in imaging bands and imaging methods. Photogrammetry is mainly to obtain accurate geographic information from remote sensing, and its application also has the above-mentioned problems. Therefore, remote sensing image acquisition technology and close-range application of photogrammetry cannot benefit from this survey.

Existing surveys on remote sensing image registration mostly include single-modal general registration frameworks and matching methods, while MMRS image registration, as a very important branch, only occupies a small part of the article [1,6–8]. A large part of the multi-modal image processing work is focused on medical image registration [9–11], and few papers specifically review the part of multi-modal images in remote sensing image processing. Common MMRS images include cross-temporal, cross-season, optical to SAR, optical to infrared, optical to Light Detection and Ranging (LIDAR), map to visible, etc. In the existing literature, research on optical-SAR [12–14] and optical-infrared [15–17] is most common. We discuss registration methods for MMRS images, rather than the types of modal image pairs. Readers who want to know about more registration methods can refer to [18]. In this regard, we review the general methods of MMRS image registration, especially classification according to the registration method category, and introduce recently popular learning-based methods, so that readers can learn about cutting-edge methods in the field at a glance. The integrated structural framework of this review is shown in Figure 1.

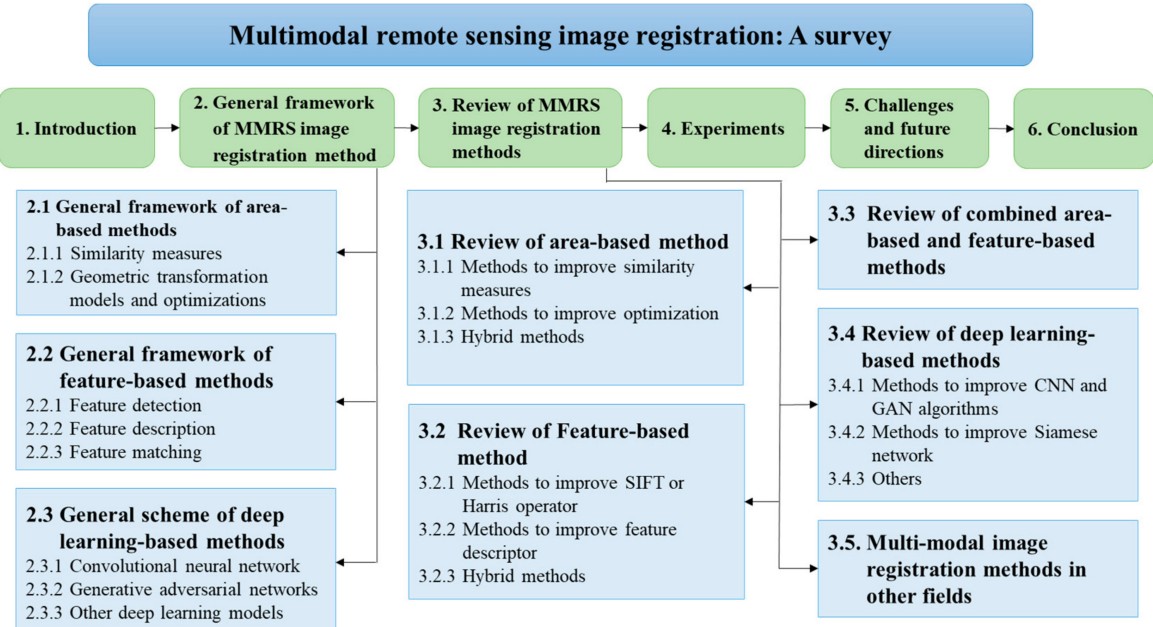

**Figure 1.** Structural framework of this review.

## 2. General Framework of MMRS Image Registration Methods

Area-based methods (also called intensity-based methods) generally use a robust similarity measure to search for the optimal geometric transformation through a predefined template window. Feature-based methods primarily include three steps: feature detection, feature description and feature matching [12].

### 2.1. General Framework of Area-Based Methods

The framework of area-based method is illustrated in Figure 2. This framework consists of three major components: namely, similarity measures, geometric transformation models and optimizations.

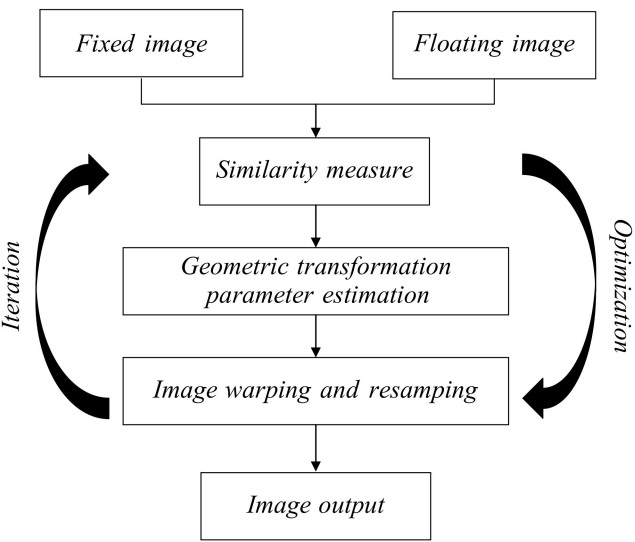

**Figure 2.** Framework of area-based image registration.

#### 2.1.1. Similarity Measures

Similarity measurement as the core factor in the iterative process of MMRS image registration is used to design various standards based on the intensity difference between two images [10]. Common similarity measures include the sum of square differences (SSD), sum of absolute differences (SAD), cross-correlation and normalized cross-correlation (NCC). Based on the assumption that two images have similar intensities, SSD directly calculates the distance between the template window and the corresponding pixels of the original image. Due to the simplicity of SSD, it has high computational efficiency. SSD can be expressed by the following formula:

$$D(x,y) = \sum_{i=1}^{m} \sum_{j=1}^{n} [I(x+i-1, y+j-1) - T(i,j)]^2 \tag{1}$$

where $T$ indicates the template window of size $m \times n$, $I$ represents the original image, in which we need to find an area matching the template. Taking $(x, y)$ in the upper left corner of the original image $I$ as the starting point, the subgraph of size $m \times n$ traverses the whole image and calculates its similarity with the template and finds the subgraph with the maximum similarity as the final matching result. NCC [19,20] judges the linear correlation of two images by calculating the correlation coefficient matrix. NCC can be expressed as:

$$\rho(x,y) = \frac{\sigma(I_{x,y}, T)}{\sqrt{D_{x,y}D}} = \frac{\frac{1}{mn} \sum_{i=1}^{m} \sum_{j=1}^{n} (I_{x,y}(i,j) - \bar{I}_{x,y})(T(i,j) - \overline{T})}{\sqrt{\frac{1}{mn} \sum_{i=1}^{m} \sum_{j=1}^{n} (I_{x,y}(i,j) - \bar{I}_{x,y})^2} \sqrt{\frac{1}{mn} \sum_{i=1}^{m} \sum_{j=1}^{n} (T(i,j) - \overline{T})^2}} \tag{2}$$

where $I_{x,y}$ indicates that the sub-block with the same size in the original image which takes $(x, y)$ as the upper left corner point. $\sigma(\cdot)$ is the covariance. $D_{x,y}$ and $D$ are the variances of $I_{x,y}$ and $T$ respectively. $\bar{I}_{x,y}$ and $\overline{T}$ are expressed as the mean gray value.

Note that SSD and NCC are sensitive to nonlinear radiation differences, they are not suitable for MMRS image registration. Mutual information (MI) is more robust to radiation

changes [21,22], whose objective, analogous to NCC, is to maximize the MI between the two images. The MI of random variables $A$ and $B$ can be expressed as [23]:

$$I(A, B) = E(A) + E(B) - E(A, B) \tag{3}$$

where $E(\cdot)$ are the entropies, and $E(A, B)$ is the joint entropy. The MI metric specifies that when $I(A, B)$ reaches its maximum value, two images are registered. The entropy and joint entropy can be calculated as:

$$E(A) = \sum_a -P_A(a) \log P_A(a) \tag{4}$$

$$E(B) = \sum_b -P_B(b) \log P_B(b) \tag{5}$$

$$E(A, B) = \sum_{a,b} -P_{A,B}(a, b) \log P_{A,B}(a, b) \tag{6}$$

where $P_A(a) = \sum_b P_{A,B}(a, b)$ and $P_B(b) = \sum_a P_{A,B}(a, b)$ are the marginal probability mass functions, and $P_{A,B}(a, b) = h(a, b)/\sum_{a,b} h(a, b)$ is the joint probability mass function, $h$ is the joint histogram of the two images, which can be represented by the following matrix:

$$h = \begin{bmatrix} h(0,0) & h(0,1) & \cdots & h(0, N-1) \\ h(1,0) & h(1,1) & \cdots & h(1, N-1) \\ \vdots & \vdots & \vdots & \vdots \\ h(M-1,0) & h(M-1,1) & \cdots & h(M-1, N-1) \end{bmatrix} \tag{7}$$

where $h(a, b)$ is the number of point pairs for which the intensity in one image is $a$ and for the other image is $b$, $M$ and $N$ are the ranges of the intensity values. We determine the MI value between two images by calculating the joint histogram of each window to be matched. Although MI is more suitable for MMRS image registration due to its robustness to nonlinear radiation differences, researchers are committed to improving this method to enhance the registration performance of MMRS by reducing the computational cost. For instance, Yang et al. [24] and Lehureau et al. [25] combined MI with feature-based methods to improve the registration accuracy of optical-to-SAR images.

### 2.1.2. Geometric Transformation Models and Optimizations

The other key steps of area-based MMRS image registration are the selection of geometric transformation models and optimizations. The transformation model, also called the mapping function, requires estimation of model parameters for image warping and resampling, and finally alignment and registration. Existing transformation models can be simply divided into linear models (for example, rigid models, affine transformation models, and projection models) and non-rigid models, such as physical models and interpolation models [10].

After selecting the metric and transformation model, it is necessary to find the optimal transformation in the iterative process of MMRS image registration to achieve optimal matching between two images. However, it is easy to fall into a local optimum during the search process, which leads to a decrease in registration performance [26]. It can be seen that the choice of the transformation model and optimization method largely determines the accuracy of registration. Readers can refer to [10,27] for more methods and details.

### 2.2. General Framework of Feature-Based Methods

Feature-based MMRS image registration includes main three steps. First, feature detection, which selects the prominent features between two images, such as point features [28], edge features [13] and region features [29]. Second, feature description, which refers to describing the extracted features for the next step of matching; and third, feature matching,

which designs a specific similarity measure for the descriptor to establish a geometric transformation model, to realize the alignment and registration between the two images. Figure 3 illustrates the feature-based framework.

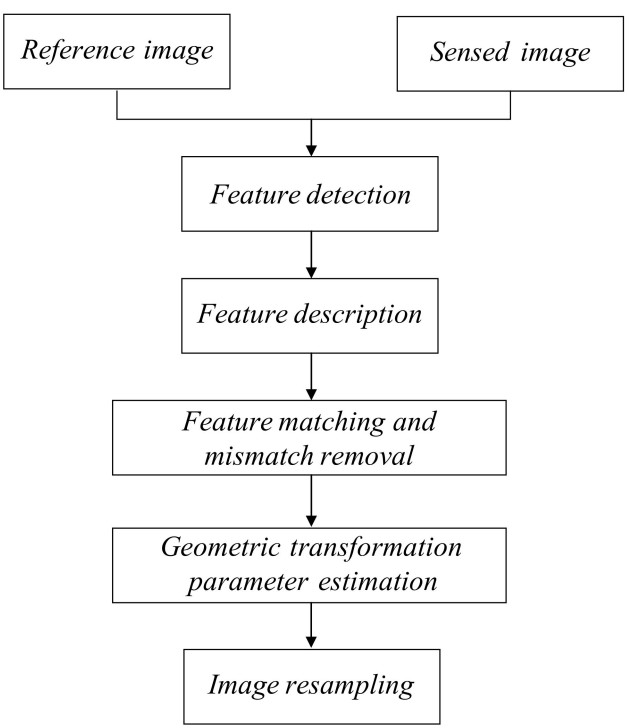

**Figure 3.** Framework of feature-based image registration.

### 2.2.1. Feature Detection

Feature detection and extraction can be divided into corner and blob detection. Corners usually refer to the position in the image where the gray value changes substantially, such as an edge intersection point. Moravec first proposed a corner detector, which mainly extracts key points from the intensity difference in a local image area [30]. After this, based on the Moravec detector, the Harris corner detector was developed by the image gradient and the response function was proposed [31]. Subsequently, the Gaussian Laplace (LOG) detector motivated Mikolajczyk [32] to combine the LOG and Harris detectors to develop a new version that locates key points in scale space, i.e., the scale-invariant Harris detection (Harris-Laplace).

Blobs generally refer to areas that are different in color and gray scale from the surrounding area (also called singular areas). The scale-invariant feature transform (SIFT) [33,34] is a classical algorithm for blob detection. Owing to its invariance to scale changes, translation and rotation, and robustness to geometric distortion, it is widely used in feature detection. The SIFT feature detector generates the key points of position and the scale by constructing a Gaussian pyramid and locating the extreme points in the difference of Gaussian (DOG) space. Due to the scale invariance and positioning accuracy of SIFT, SIFT-like methods have been widely used in various applications. Many academics have successfully improved SIFT to eliminate multi-modal difference, thereby achieving MMRS image registration, including multispectral [35,36], optical-to-SAR images [5,14] and visible-to-infrared images [37].

### 2.2.2. Feature Description

Feature description refers to generating specific descriptors for the extracted feature points to prepare for the follow-up feature matching. This step directly determines the registration performance, which requires generating stable descriptors for the matched features and being robust to geometric transformation and image resolution. SIFT men-

tioned above and SURF (speeded up robust features) [38] are the two most common feature descriptors. The major idea is to calculate the gradient amplitude and direction around the interest point to generate a histogram descriptor. At the same time, Gaussian derivative, moment invariant, and shape context are also frequently used descriptors.

### 2.2.3. Feature Matching

Assuming that two sets of interest points have been extracted from the reference image and the sensed image, feature matching finds the correspondence between them based on a specific algorithm to obtain more matching point pairs. A direct method is to correspond the two interest points by directly using spatial geometrical relations like graph matching and point set registration. An indirect method is based on the similarity of local feature descriptors with false matches being rejected using local and/or global geometrical constraints. The most typical methods for mismatch removal are resampling-based methods, which are also known as fast sample consensus (FSC) and random sample consensus (RANSAC) [39].

### 2.3. General Framework of Deep Learning-Based Methods

Traditional image registration technology is constrained by image resolution, intensity differences and low computational efficiency. With the rapid development of deep learning, learning-based methods can promote an iterative area-based method, directly estimate the geometric transformation parameters, and reduce the computational complexity. At the same time, in the process of MMRS image registration, deep learning shows great potential in detecting feature points of images with appearance differences, further improving the registration performance. Common methods for solving MMRS image registration problems based on deep learning are described in detail in Section 3.4.

### 2.3.1. Convolutional Neural Network

The model construction of deep learning is based on a neural network and consists of many layers [40]. These layers transform input data into output by learning features. The "hidden layers" are usually between the input and output layers. Deep learning refers to a large number of hidden layers [40]. Convolutional neural network (CNN) is one of the most common models in the field of deep learning. Because of the structure characteristic of continuous layers, it can capture more complex image features and learn features for registration tasks. These continuous layers are specifically divided into convolutional layer, pooling layer and fully connected layer. The convolution operation is mainly to extract image features. With the increase of convolutional layers, multi-layer networks can extract richer image features. After convolution, there are still many dimensional features of the image. The pooling layer divides the feature matrix into several individual blocks and takes the maximum or average value; that is, maximum pooling and average pooling, which play a role in dimensionality reduction. Finally, the fully connected layer non-linearly combines all the local features and the feature matrix of each channel to obtain the output.

### 2.3.2. Generative Adversarial Networks

At present, Generative adversarial networks (GANs) [41] is a very popular technology for deep learning. It is composed of the generator and the discriminator. In the learning process, the generator makes the output image as real as possible, while the discriminator has to work hard to identify the true and false images. This process is similar to a "two-person game". Both networks try to optimize two completely opposite loss functions [42]. Unlike CNN's powerful ability to analyze data and extract features, GANs focus on generating data, enhancing data by the adversarial network, or generating fake images to eliminate modal differences.

### 2.3.3. Other Deep Learning Models

Widely used deep learning models include recurrent neural networks (RNN) [43], autoencoders (AEs) [44], stacked autoencoders and Restricted Boltzmann machines [45]. They are mostly utilized in the fields of sequence analysis [46], feature representation [47,48], data compression and dimensionality reduction [49], and image classification [50–53], and do not involve remote sensing image registration.

## 3. Review of MMRS Image Registration Methods

Remote sensing images are obtained from remote sensing satellites, which record the magnitude of electromagnetic waves of various ground objects. Depending on the imaging system, it mainly includes optical, SAR, Light Detection and Ranging (LIDAR) and Radio Detection and Ranging (RADAR) data, which could be combined for remote sensing applications. For MMRS image registration, there are several types of multi-source, multi-spectrum, and multi-temporal algorithms. Common research mainly includes the following types: optical-to-SAR, optical-to-LIDAR, infrared-to-visual, and visual-to-map images. As shown in Figure 4, in the optical and SAR images of the same area, due to the differences among imaging systems and the presence of speckle noise, it is very difficult to make use of traditional methods to register two images.

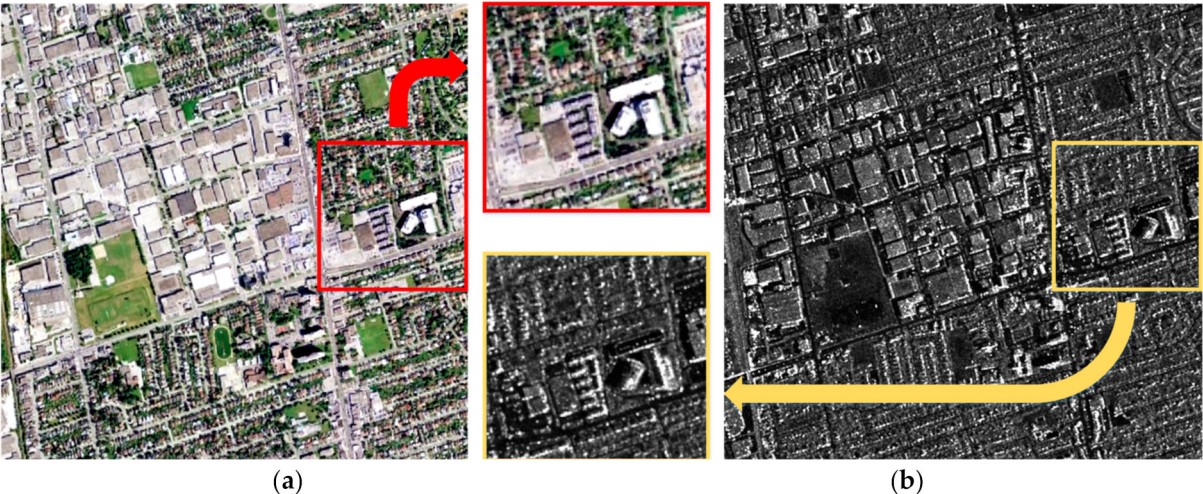

(**a**)          (**b**)

**Figure 4.** An example of radiation difference between MMRS images: (**a**) Optical image. (**b**) SAR image. As shown by the enlarged patch, the radiation differences and noise effects are particularly noticeable.

For this type of MMRS image registration problem, with significant non-linear radiation differences, the area-based method is susceptible to problems of image overlap, geometric distortion, and high-resolution images with high computational cost. However, the typical feature-based method (SIFT) is not suitable for multi-modal images. It is difficult to extract a large number of matching point pairs and find a suitable descriptor to match them. In order to improve the accuracy and efficiency of MMRS image registration, researchers have proposed many improvements. This section will review and discuss these methods in detail. The following mainly include area-based, feature-based and deep learning-based improvements.

### 3.1. Review of Area-Based Methods

As mentioned in Section 2, an area-based method finds an appropriate similarity measure in two images to accurately estimate the optimal parameters of the geometric transformation in the iterative process. However, this method needs a large amount of computation and is easily trapped into a local optima. Thus, relevant researchers have made improvements in the following aspects:

1.  Improving the similarity measure MI that is the most suitable for MMRS image registration;
2.  Establishing a similarity measure to eliminate modal differences based on the shape and structure of an image;
3.  Improving the optimization method for high-resolution and noise-affected images to enhance registration performance.

Table 1 shows a summary of the improvements on area-based algorithms in MMRS image registration.

**Table 1.** Analysis of area-based method improvements.

| Author | Method Category | Method Improvement | Conclusion |
|---|---|---|---|
| Cole-Rhodes et al. [21] | Methods to improve similarity measures | Improve the joint histogram of MI and propose the simultaneous perturbation stochastic approximation (SPSA) algorithm. | The algorithm relies on gradient approximation instead of the target gradient value and finds the global maximum value through the local extrema of the objective function, which greatly speeds up the entire registration process. |
| Chen et al. [23] | Methods to improve similarity measures | A new joint histogram estimation algorithm is proposed, called Generalized Partial Volume Estimation (GPVE, with a second-order B-spline function). | Overcome the problem of interpolation-induced artifacts. |
| Shadaydeh et al. [54] | Methods to improve similarity measures | Propose a weight-based joint histogram estimation method (WJH-MI). | Reduces the peaks in the joint histogram caused by the background or homogeneous regions by assigning more weight to the high gradient pixels containing more registration related information. |
| Xu et al. [55] | Methods to improve similarity measures | Propose using the symmetric form of Kullback-Leiber divergence, namely Jeffrey's divergence as similarity measure. | The registration model based on Jeffrey's divergence can provide a larger feasible search space and solve the problem of MI facing insufficient image overlapping area. |
| Xie et al. [56] | Methods to improve similarity measures | MLPC method combining multi-scale Log-Gabor filter and phase consistency | Effectively resolving the non-linear intensity difference in MMRS image registration. |
| Xie et al. [57] | Methods to improve similarity measures | Based on the extended phase correlation of log-Gabor, an improved LGEPC method is proposed | Use the overall structure information to eliminate the influence of radiation differences as much as possible. |
| Hasan et al. [58] | Methods to improve optimization | Using CCRE to align SAR and Google satellite images and applying partial volume interpolation to calculate the gradient of the similarity measure. | Directly implement the optimization process based on partial volume interpolation. |
| Dame and Marchand [59] | Methods to improve optimization | Define a new inverse combinatorial optimization method to handle the quasi-concave shape of MI, where the required derivative can be pre-calculated allowing the Hessian matrix to be estimated after convergence. | Reduce the calculation time and estimate an accurate parameter. |
| Liang et al. [60] and Wu et al. [61] | Methods to improve optimization | The authors used the ant colony optimization (ACO) algorithm to optimize the similarity measure to maximize MI. | The similarity curve of MI has been proved to have many local optimal values. |

**Table 1.** *Cont.*

| Author | Method Category | Method Improvement | Conclusion |
|---|---|---|---|
| Yan et al. [26] | Methods to improve optimization | Propose a method of using transfer optimization (TO) to maximize MI, optimize conversion parameters and transfer better results to another optimizer in the iterative process. | Enhance the global search capability of TO and avoid a local optima. |
| Liu et al. [62] | Hybrid methods | Propose an image registration method that combines local self-similarity (LSS) and MI, which combined local internal features and global intensity information. | The proposed method can register multi-sensor images with different resolutions and handle the geometric differences between grayscale and corresponding pixels and regions. |
| Ye et al. [63] | Hybrid methods | A fast and robust template matching framework for MMRS images called channel features of orientated gradients (CFOG). | A new general scheme of template matching based on pixel feature representation, suitable for all kinds of multi-modal image registration. |
| Yan et al. [2] | Hybrid methods | Introduced the histogram of oriented gradient distance (HOGD) and the grey wolf optimizer (GWO). | It avoids falling into the local optimum and reduces the calculation time. |
| Liang et al. [64] | Hybrid methods | A variable template matching method based on DOG features and the sum of cosine difference. | The proposed matching method has good robustness to nonlinear light intensity changes, and effectively improves the matching accuracy. |

### 3.1.1. Methods to Improve Similarity Measures

Multi-resolution remote sensing image registration using stochastic gradient to optimize the MI similarity metric was proposed in [21]. First, it improved the joint histogram of MI into 64 bins to produce a smoother surface, which can better integrate the optimization method. In addition, the author proposed using the simultaneous perturbation stochastic approximation (SPSA) algorithm, which relies on an efficient gradient approximation instead of an accurate target gradient value. Then, the global maximum value was found through a local extrema of the objective function, which greatly accelerates the entire registration process. Chen et al. [23] put forward a new joint histogram estimation algorithm called generalized partial volume estimation (GPVE, with a second-order B-spline function) for computing MI to register multi-temporal remote sensing images, to overcome the problem of interpolation-induced artifacts.

Regardless of whether mutual information (MI) or normalized mutual information (NMI) overcomes the overlapping area problem, the estimation of the similarity measure requires calculating its joint histogram. Shadaydeh et al. [54] proposed a weight-based joint histogram estimation method (WJH-MI). That is, each bin in the joint intensity histogram is calculated as the sum of the weights corresponding to the pixel intensity value of the bin. The weight of each pixel is defined as the exponential function of the distance image and the normalized gradient image. This method reduces the peaks in the joint histogram caused by the background or homogeneous regions by assigning more weight to the high gradient pixels containing more registration related information. Experiments show that the proposed method produces a better similarity measurement surface, with more obvious peaks and fewer registration errors. In order to solve the problem of MI (also known as Kullback–Leiber divergence) facing insufficient image overlapping area, Xu et al. [55] proposed using the symmetric form of Kullback–Leiber divergence, namely Jeffrey's divergence as similarity measure. Jeffrey's divergence quantifies the distinction of image pairs by bidirectionally measuring the "distance" between the joint histogram and the product edge histogram of two images. Derived from the definition, the registration model based on Jeffrey's divergence can provide a larger feasible search space. Experiments

showed that the sensitivity of Jeffrey's divergence to scene overlap is lower than other measures, and it is more suitable for registering small size images.

Structure and shape features have been used as similarity measures in multi-modal medical image registration, achieving better performance than traditional similarity measures. The features extracted by the phase correlation or phase congruency (PC) algorithm proposed by De Castro [65] and Reddy [66] are widely used in robust MMRS image registration due to their insensitivity to changes in illumination and contrast. Xie et al. [56] proposed the MLPC method, based on the multi-scale Log-Gabor [67] filter combined with traditional phase congruency methods [65,66], effectively resolving the non-linear intensity difference in MMRS image registration. The authors obtained the amplitude of the reference and sensed images in the frequency domain after Fourier transform, and then used log-Gabor filters with different central frequencies to filter the amplitude to obtain a series of filtered images. After filtering, a multi-scale map space was constructed and each image pair in the map was phase-correlated. The maximum response peak was considered as the optimal solution, and the coordinates of the maximum response peak were used to obtain the rotation angle and scale factor. Finally, the cross-power spectrum between the superimposed structure spectrum of the reference image and the corrected sensed image was calculated to eliminate the radiation difference. The inverse Fourier transform was used to obtain the maximum peak value, and the coordinates of the maximum peak value determined the transformation. Subsequently, based on this article, the authors proposed an improved LGEPC [57], which is an extended phase correlation algorithm based on log-Gabor. Similarly, the filtered amplitude image obtained by log-Gabor filters with different central frequencies was used to construct a multi-scale atlas space and obtain rotation and scale factors. In addition, the filter structure spectrum obtained by log-Gabor filters of different central frequencies was superimposed together to enhance the overall structure information, which helps eliminate the influence of radiation differences as much as possible on the step of solving translation in the extended phase correlation.

### 3.1.2. Methods to Improve Optimization

A new similarity measure, called cross-cumulative residual entropy (CCRE), was successfully applied to multi-modal medical image registration to adapt to images with different brightness and contrast, while being more robust to noise. Hasan et al. [58] used CCRE to align SAR and Google satellite images, and at the same time extended the Parson-window optimization method proposed by Thevenaz [68]. The authors applied partial volume interpolation to calculate the gradient of the similarity measure instead of the joint histogram, which allowed the authors to directly implement the optimization process based on partial volume interpolation. Results showed that the use of partial volume interpolation in the optimization process significantly improved the registration success rate and accuracy of CCRE-based and MI-based algorithms. The authors defined a new inverse combinatorial optimization method in [59] to handle the quasi-concave shape of MI, where the required derivative can be pre-calculated allowing the Hessian matrix to be estimated after convergence. Thus, this method can reduce the calculation time and estimate an accurate parameter. Note that the definition of MI has been adapted to the differential image alignment problem, so that the alignment function is as smooth and concave as possible and retains robustness to multi-modal image intensity changes. In addition, utilizing a new method based on reference pixel selection greatly reduces time consumption, which results in an accurate, fast and robust registration process. In [60,61], the authors used the ant colony optimization (ACO) algorithm to optimize the similarity measure to maximize MI. However, the similarity curve of MI has been proved to have many local optimal values. Thus, Yan et al. [26] was inspired by transfer learning and proposed a method of using transfer optimization (TO) to maximize MI, optimize conversion parameters and transfer better results to another optimizer in the iterative process. This helps enhance the global search capability of TO and avoid a local optima.

3.1.3. Hybrid Methods

Liu et al. [62] proposed an image registration method that combines local self-similarity (LSS) and MI, which combined local internal features and global intensity information to achieve a better registration effect. A likelihood map was calculated for each CP in the image using the Gaussian pyramid weighted Bayesian probabilistic model. The position of the peak in the likelihood map represented the corresponding CPs in the image. At the same time, the authors took the maximum of MI as the optimization goal and introduced the particle swarm optimization (PSO) algorithm to search for the best parameter of the geometric mapping function. Considering effectiveness and performance, experiments show that the proposed method can register multi-sensor images with different resolutions and handle the geometric differences between grayscale and corresponding pixels and regions. Ye et al. [63] proposed a fast and robust template matching framework for MMRS images. The first step extracts a local descriptor at each pixel of the image, which can be HOG, LSS, or SURF to generate a pixelized feature representation. Here, the author introduced a novel pixelized feature representation called channel features of orientated gradients (CFOG), which is an extension of the pixelated HOG descriptor. The pixelated feature representation can be constructed by using HOG descriptors with a single unit block. The trilinear interpolation in a single unit block can be regarded as a convolution operation with a triangular kernel. Thus, the HOG descriptor was reconstructed by convolution in the image gradient of a specific direction, and the convolution was performed by the Gaussian kernel instead of the triangular kernel. The reconstructed CFOG descriptor suppressed noise more effectively and reduced the contribution of the gradient away from the center of a region, improving the computational efficiency. Subsequently, a new template matching similarity measure based on pixel feature representation was introduced, and fast Fourier transform was used for faster computation. Yan et al. [2] proposed a new similarity measure, the histogram of oriented gradient distance (HOGD). In order to avoid falling into local optimality, the grey wolf optimizer (GWO) was introduced since it exhibited global search capabilities for complex optimization problems. GWO searched for optimal transformation parameters by minimizing HOGD. To reduce computation time, GWO was combined with a data-driven strategy, namely DDGWO. In DDGWO, a support vector machine (SVM) model was trained to predict HOGD instead of calculating it directly, which can lead to a significant reduction in computation time. Liang et al. [64] proposed a fast-matching method based on dominant orientation of gradient (DOG), which constructed a feature map by extracting the DOG feature of each pixel in an image. The author defined a new similarity measure called the sum of cosine difference, which can be accelerated by the Fast Fourier Transform (FFT). In order to improve the matching performance, a new variable template matching (VTM) method was proposed to determine the correspondence between images. Experimental results showed that the proposed matching method had good robustness to nonlinear light intensity changes, and effectively improved the matching accuracy.

*3.2. Review of Feature-Based Methods*

Researchers have made many improvements and contributions to the area-based methods. However, there are still cumbersome and time-consuming calculations in the iterative optimization process caused by high resolution and strong noise. Most methods rely on geometric correction based on the geographic information of an image. However, when geographic information is not available, an image is difficult to optimize considering large geometric changes. Feature-based methods are robust to rotation, translation and geometric distortion; but, due to speckle noise and intensity differences, it is very difficult to extract feature points between two images and it is easy to remove outliers. Hence, how to detect reasonable features between reference and sensed images and design suitable descriptors for them has become a challenge for scholars. Table 2 shows a summary of the improvements on feature-based algorithms in MMRS image registration.

**Table 2.** Analysis of feature-based method improvements.

| Author | Method Category | Method Improvement | Conclusion |
|---|---|---|---|
| Yu et al. [28] | Methods to improve SIFT or Harris operator | Combine the advantages of SIFT and Harris to propose a coarse to fine MMRS image registration. | The proposed method is more suitable for MMRS image registration. |
| Sedaghat et al. [35] | Methods to improve SIFT or Harris operator | Propose a registration method, called Uniform Robust SIFT (UR-SIFT), suitable for a variety of optical multi-source remote sensing images with illumination, rotation and up to five times the scale difference. | In the process of descriptor merging, this algorithm led to many other combinations to be merged into the same final descriptor, which greatly reduced the registration accuracy. |
| Hossain et al. [69] | Methods to improve SIFT or Harris operator | An improved symmetric SIFT method. | The descriptor merging step in symmetric SIFT can be skipped to obtain excellent matching accuracy. |
| Fan et al. [5] | Methods to improve SIFT or Harris operator | A novel spatial consensus matching (SCM) algorithm. | Using improved SIFT and K-nearest neighbors (KNN) to obtain an initial set of matching features and utilized spatial consistency constraints for refinement. |
| Huang et al. [70] | Methods to improve SIFT or Harris operator | A new algorithm for improving feature extraction and feature matching was introduced. | Using Harris operator, Canny operator and shape context descriptor to match multimodal images. |
| Xiang et al. [12] | Methods to improve SIFT or Harris operator | Propose a SIFT-like algorithm (OS-SIFT), which introduce multi-scale ratio of exponentially weighted averages (ROEWA) and multi-scale Sobel operators. | By calculating the consistent gradients of SAR and optical images, it is proved that the algorithm is robust to noise and has excellent registration performance and accuracy. |
| Aguilera et al. [71] | Methods to improve feature descriptor | The edge-oriented histogram (EOH) descriptor | The descriptor contained the information of the contour near each feature point without using the gradient information to describe the shape and contour of the image. |
| Ye et al. [72] | Methods to improve feature descriptor | Combine Harris and LSS descriptors to establish a piecewise linear transformation | The impact of the low discriminability of the LSS descriptor is reduced, and the reliable registration of multi-spectral remote sensing images is realized. |
| Sedaghat et al. [73] | Methods to improve feature descriptor | An advanced version of the self-similarity descriptor, which has high distinguishability, called distinctive order based self similarity (DOBSS) descriptor. | The DOBSS descriptor has better recall, precision and positioning accuracy. |
| Ye et al. [74] | Methods to improve feature descriptor | Based on the internal self-similarity of images, the author introduced a shape descriptor of dense local self-similarity (DLSS). | Using the internal structure information of the image to construct the descriptor is more suitable for multi-mode image registration. |
| Xiong et al. [75] | Methods to improve feature descriptor | A rank-based local self-similarity (RLSS) to describe the local shape of an image in a distinguishable manner. | The rank value was used as a substitute for the correlation value to indicate the relative relationship of the correlation value, which further improved distinguishability of feature descriptors. |
| Cui et al. [76] | Methods to improve feature descriptor | A multi-scale phase-congruency descriptor (MS-PC), which captured the shape and structural characteristics of an image. | The descriptor compensates for the sensitivity of traditional descriptors to radiation differences. |

**Table 2.** *Cont.*

| Author | Method Category | Method Improvement | Conclusion |
|---|---|---|---|
| Ye et al. [4] | Methods to improve feature descriptor | Combine the PC describing structural features with the histogram of gradient directions (HOG) as a feature descriptor, called HOPC. | Using the structure information of the image to construct the descriptor can eliminate modal differences and is more suitable for multi-mode image registration. |
| Liu et al. [15] | Methods to improve feature descriptor | A maximum stable phase congruency (MSPC), which combined affine invariant region extraction and image structural features. | The algorithm extracted structural features by merging phase congruency images in multiple orientations. Registration was achieved according to the correspondence of the descriptors. |
| Ye et al. [77] | Hybrid methods | A local invariant feature that was robust to geometric distortion and radiation changes, which consisted of a feature detector named MMPC-lap and a feature descriptor named local histogram of orientated phase congruency (LHOPC). | This method also solved the radiation differences caused by spectral and time changes between MMRS images. |
| Fan et al. [78] | Hybrid methods | An UND-Harris detector that introduced nonlinear diffusion, feature ratio and block strategy. | Experimental results on different SAR and optical image pairs showed the effectiveness of this method, which can obtain better registration results and improve registration accuracy. |
| Zeng et al. [37] | Hybrid methods | An infrared-to-visible image registration method based on morphological gradient and C-SIFT. | The algorithm uses morphological methods to preserve the gray-scale edges of the image and improve the similarity of infrared and visible images. |
| Li et al. [79] | Hybrid methods | A radiation insensitive image registration method based on phase congruency (PC) and a maximum index image (MIM), which was called radiation variation insensitive feature transform (RIFT). | This method realizes the insensitivity and rotation invariance to multi-modal image radiation changes. |
| Sui et al. [13] | Hybrid methods | An iterative process combining line segment extraction and line intersection matching based online segment extraction, and integrated Voronoi polygons into spectral point matching (SPM) to obtain the correspondences between line intersections. | An iterative strategy of "re-extraction" and "re-matching" mechanisms was adopted to enhance feature extraction and matching performance. |
| Zhao et al. [80] | Hybrid methods | Using Kovesi corner point extraction and line segment detection methods based on phase congruency and local direction. | Compared to other edge extraction methods, this method extracted more equivalent line segments. |
| Xu et al. [81] | Hybrid methods | A new contour segment representation method based on local histogram of maximal edge orientation and defined angles is proposed, and the Fréchet distance is defined as the weighting parameter of combined histograms to enhance the descriptive ability. | The proposed method can effectively reduce the impacts of radiation distortion and is superior to some current popular multi-source image matching methods. |

### 3.2.1. Methods to Improve SIFT or Harris Operator

SIFT and Harris are relatively mature and complete image registration methods. Therefore, researchers have improved on both methods directly, considering MMRS image registration. In [28], the authors combined the advantages of SIFT and Harris to propose a coarse to fine MMRS image registration. First, SIFT was used to automatically find the CPs between the images to roughly match the reference and sensed images. Then, the affine transformation model was utilized to detect the transformation. Next, the algorithm applied Harris detector to perform fine registration on the coarsely matched image while processing the local deformation in the image. Then, a set of dense feature points from the input image were found. After that, the wavelet pyramid was used to search for the corresponding connection points of these feature points. Simultaneously, the error in matching was eliminated by the global consistency check method. Finally, the input image was corrected by piecewise linear transformations and the triangular irregular network (TIN). In [35], the authors proposed a registration method, called Uniform Robust SIFT (UR-SIFT), suitable for a variety of optical multi-source remote sensing images with illumination, rotation and up to five times the scale difference. A symmetric SIFT [82] algorithm adapted to multimodal invariance was widely used in MMRS image registration. However, in the process of descriptor merging, this algorithm led to many other combinations to be merged into the same final descriptor, which greatly reduced the registration accuracy. In order to address the above problems, Hossain et al. [69] proposed an improved symmetric SIFT method. The overall direction difference between the given images was estimated by analysing the initial matching set, and the estimated value was used to rotate and normalize the descriptor in the second step. This way of normalization has not caused area reversal. Hence, the descriptor merging step in symmetric SIFT can be skipped to obtain excellent matching accuracy.

Fan et al. [5] proposed a novel spatial consensus matching (SCM) algorithm, which used improved SIFT and K-nearest neighbors (KNN) to obtain an initial set of matching features and utilized spatial consistency constraints (the low distortion constraint was used) for refinement. The algorithm obtained matching features by gradually adding features that are spatially consistent with the currently obtained matching features. Finally, RANSAC was used to estimate the transformation parameters from spatially consistent matching features. In [70], a new algorithm for improving feature extraction and feature matching was introduced. The Harris operator was utilized to extract the CPs of the reference image, and the Canny operator was used to extract the edge features of the reference and sensed images, while processed by the dilation algorithm to retain useful edge information. Finally, the improved shape context descriptor was applied to match the image by comparing the edge feature distribution in the defined circular template.

Xiang et al. [12] proposed a SIFT-like algorithm (OS-SIFT) in order to achieve high-resolution optical-to-SAR image registration. Experimental results on simulated images and multiple high-resolution satellite images showed that the algorithm was excellent in registration performance and accuracy. Two different operators were introduced in this algorithm, multi-scale ratio of exponentially weighted averages (ROEWA) [83] and multi-scale Sobel, to calculate the consistent gradient of SAR and optical images to demonstrate robustness to noise. Then, two Harris scale spaces were constructed for SAR and optical images, and a location refinement method based on key point spatial information was proposed. That is, key points of different scales belonging to the same corner point should have similar structural properties, which means they were in a straight line. Finally, orientation restrictions and multiple image blocks were used to construct GLOH-like descriptors, which contained more structural information and produced more stable registration results. In successive research, Xiang et al. proposed a dense registration algorithm that combines robust features and optical flow, namely OS-flow [84]. First, this algorithm extracted two dense feature descriptors for each pixel in the optical and SAR images, i.e., the optical GLOH descriptor and the SAR GLOH descriptor. Using dense descriptors instead of brightness values can satisfy the assumption of brightness constancy. Then, the global and local

methods were constructed. The global method estimated the flow graph by optimizing the objective function, and the local method iteratively estimated the flow vector in the local neighborhood. Finally, a coarse-to-fine matching strategy was adopted to handle large displacements and improve efficiency. In [85], the authors proposed a subpixel registration method that combined robust features and 3-D PC (phase correlation). Robust features captured the inherent properties of two images and retained their structural information. 3-D PC used image cubes as a substitute for the two original images, which estimated the 2-D translation by locating the peak in the spatial domain or directly in the Fourier domain. In addition, in order to improve the accuracy of 3-D PC, a constrained energy minimization algorithm was introduced, which was used to find the Dirac delta function after the inverse Fourier transform, as well as fast sample consistency fitting to estimate the phase difference after high-order singular value decomposition of the PC matrix.

### 3.2.2. Methods to Improve Feature Descriptor

Designing a robust and effective descriptor for MMRS images has become the most popular improvement method nowadays. In order to avoid the lack of correlation of gradients caused by nonlinear intensity differences, Aguilera et al. [71] proposed the edge-oriented histogram (EOH) descriptor to merge the spatial information of the contour from each key point without using the gradient information. The descriptor contained the information of the contour near each feature point to describe the shape and contour of the image. Finally, a large number of multispectral image pairs were used to evaluate the proposed method.

Local self-similarity (LSS) [86] is a local feature descriptor that captures the internal geometric structure of the image based on a grid of log-polar coordinates, which is insensitive to color and intensity changes. Therefore, in [72], the authors used SR-SIFT with scale limitation to first detect key points, then they combined Harris and LSS descriptors to establish a piecewise linear transformation to achieve reliable registration of multi-spectral remote sensing images. However, the motivation for this method is to minimize the impact of the low distinguishability of the LSS descriptor. In general, a good descriptor should be robust and distinctive. In order to satisfy the above properties, Sedaghat et al. [73] proposed an advanced version of the self-similarity descriptor, which has high distinguishability, called distinctive order based self-similarity (DOBSS) descriptor. First, the UR-SIFT proposed by the author's previous work was used to extract a set of uniform and dense features. Second, the authors adopted a correlation value-based direction assignment method to construct descriptors, and at the same time utilized correlation values to group pixels in a local area to improve the separability of the descriptors. Finally, all the log-polar coordinate calculation descriptors were connected to form the final descriptor. Experiments showed that the DOBSS descriptor has better recall, precision and positioning accuracy. Based on the internal self-similarity of images, Ye et al. [74] introduced a shape descriptor of dense local self-similarity (DLSS). First, this method selected a template window on the image and divided the window into spatial regions, called "cells" containing pixels. Then, it extracts and normalizes the LSS descriptor of each cell and passes through a dense overlapping network that covers the template window. The LSS descriptors collected by all the cells in the grid were combined to construct a DLSS. Finally, the normalized cross-correlation of DLSS descriptors was used to define a similarity measure (called DLSC), and a template matching strategy was utilized to detect the correspondence between images. Based on DLSS and inspired by the Spearman's rank correlation coefficient in statistics, Xiong et al. [75] proposed rank-based local self-similarity (RLSS) to describe the local shape of an image in a distinguishable manner. The rank value was used as a substitute for the correlation value to indicate the relative relationship of the correlation value. In addition, similar to the scheme of Ye et al., this descriptor was integrated into a dense sampling grid to obtain a dense RLSS descriptor (DRLSS), which further improved distinguishability of feature descriptors.

Another popular approach is to use PC to describe the detected key points in the frequency domain and design efficient descriptors. Cui et al. [76] proposed a multi-scale phase-congruency descriptor (MS-PC), which captured the shape and structural characteristics of an image to compensate for the sensitivity of traditional descriptors to radiation differences. First, the extreme values were detected in the DOG space, the log-Gabor wavelet was used to construct the phase congruency map, and the MS-PC feature vector was established. Finally, the Euclidean distance was introduced as the matching metric to obtain the final reliable matching point pair. Ye et al. [4] were inspired by medical image processing and combined the PC describing structural features with the histogram of gradient directions (HOG) as a feature descriptor, called HOPC. The author first detected the CPs through the block-based Harris operator, at the same time sorted the Harris values in each block from large to small and selected the first k points as interest points. Then, they defined a similarity measure named according to the NCC of the HOPC descriptor and used the fast template matching scheme to match the control points. Mismatches were eliminated by the global constraints in the projection transformation model, and ultimately estimated by a piecewise linear model to achieve precise registration. The authors verified the robustness and accuracy of the proposed method on multiple sets of MMRS images. In [15], Liu et al. proposed a method of registering infrared and visible images, namely maximum stable phase congruency (MSPC), which combined affine invariant region extraction and image structural features. They used moment ranking analysis to detect feature points and extracted structural features by merging phase congruency images in multiple orientations. After obtaining the neighborhood centered on the feature point through the log-Gabor filter response, the maximum stable extreme value region (MSER) was used to determine the affine invariant region of the feature point. Finally, the oriented phase congruency was constructed from these regions to describe the structural features, and registration was achieved according to the correspondence of the descriptors.

### 3.2.3. Hybrid Methods

Traditional methods are often unsatisfactory in extracting reliable matching point pairs, so they cannot generate robust descriptors to estimate geometric transformations accurately. Based on the feature-based registration method, researchers have improved both feature point extraction and feature description to achieve MMRS image registration that is more adaptable to nonlinear intensity differences.

For solving the problem of local invariant features being sensitive to the radiation difference between multi-sensor images, Ye et al. [77] proposed a local invariant feature that was robust to geometric distortion and radiation changes. It consisted of a feature detector named MMPC-lap and a feature descriptor named local histogram of orientated phase congruency (LHOPC). Inspired by Harris-Laplace, MMPC-lap combined the minimum moment of phase congruency with LOG, used the MMPC to detect key points, and then utilized LOG for scale location. Once a set of key points was detected, the phase congruency magnitudes and orientations were applied to construct the descriptor according to the spatial arrangement of DAISY [87]. In addition to adapting to geometric distortions such as scale and rotation, this method also solved the radiation differences caused by spectral and time changes between MMRS images. Compared to Gaussian smoothing (GS), nonlinear diffusion [88,89] can better preserve edge features and details while suppressing noise. At the same time, previous experiments also show that uniformly distributed Harris corners are more reliable in the matching process. Thus, Fan et al. [78] proposed an UND-Harris detector that introduced nonlinear diffusion, feature ratio and block strategy. Secondly, the phase congruency structure descriptor (PCSD) was inspired by the LSS descriptor and was built on the PC structure image of each extracted point in the form of PC sequential grouping. Experimental results on different SAR and optical image pairs showed the effectiveness of this method, which can obtain better registration results and improve registration accuracy.

The greatest challenge for the registration of infrared and visible images is the grayscale difference between the two images, but they both retain obvious contour edges. Therefore, morphological methods can be used to retain the grayscale edges of the image and improve the similarity. Consequently, Zeng et al. [37] proposed an infrared-to-visible image registration method based on morphological gradient and C-SIFT. First, the rough edges of the image were extracted by grayscale morphology. Using structural elements as a template, the authors retrieved the maximum (minimum) gray value of the template size in the image to obtain the morphologically dilated (eroded) image; thereby obtaining the gray morphological gradient image, which contained the rough edges in the image. In C-SIFT, DOG was used to extract feature points to ensure scale invariance, and the centroid method was used to obtain the main directions of feature points. In addition, BRIEF feature descriptors with fast calculation speed and strong gray invariance were introduced, and Hamming distance was utilized as a similarity measure to match feature points. Through in-depth research on phase congruency, Li et al. [79] proposed a radiation insensitive image registration method based on phase congruency (PC) and a maximum index image (MIM), which was called radiation variation insensitive feature transform (RIFT). First, the precise edge map, i.e., the PC map, was used to obtain the minimum moment map corner features and the maximum moment map edge features, to obtain a large number of repeatable features. Then, the algorithm employed the MIM based on the log-Gabor convolution sequence to describe the feature and constructed multiple MIMs to realize the rotation invariance by analyzing the internal influence of the rotation on the MIM value. The authors verified the reliability and accuracy of RIFT in the registration process through multiple qualitative and quantitative comparisons.

LSD [90] is a line segment detection algorithm, which can obtain high-precision line segment detection results in a short time. The LSD line detection algorithm first calculates the gradient size and direction of all points in the image, and then takes the adjacent points with small gradient direction changes as a connected domain. Subsequently, according to the rectangular degree of each domain, we need to decide whether to disconnect it to form a plurality of domains with a larger rectangular degree. Finally, all the generated domains are improved and filtered, and the domains that meet the conditions are retained as the final straight line detection result. The advantage of this algorithm is that the detection speed is fast, there is no need for parameter adjustment, and the error control method is used to improve the accuracy of straight-line detection. Sui et al. [13] proposed an iterative process combining line segment extraction and line intersection matching based online segment extraction, and integrated Voronoi polygons into spectral point matching (SPM) to obtain the correspondences between line intersections. At the same time, an iterative strategy of "re-extraction" and "re-matching" mechanisms was adopted to enhance feature extraction and matching performance. Zhao et al. [80] used Kovesi corner point extraction and line segment detection methods based on phase congruency and local direction. Compared to other edge extraction methods, this method extracted more equivalent line segments. Then, a new multi-modal robust line segment descriptor (MRLSD) was proposed, which was calculated by using line segment information located in a circular feature region. Based on this, the MRLSD matching method was presented. Xu et al. [81] proposed a combination of feature descriptors called "Angle Histogram and Maximum Edge Orientation Distribution" (HAED). First, they used the angle and edge direction distribution to extract the image information to generate contour segment features and capture the local and global textures respectively. Second, similarity was calculated using the Fréchet distance metric between the curves, which is the weight parameter of the histogram of each contour segment. Finally, the precise bilateral matching rules were used to match the corresponding contour segments. This method can effectively reduce the influence of radiation distortion and is better than current popular multi-source image matching methods.

### 3.3. Review of Combined Area-Based and Feature-Based Methods

Template matching and feature-based methods have their own advantages and disadvantages in MMRS image registration. By extracting reliable matches between two images and finding the corresponding relationship between them, they can handle rotation, translation, scale difference and geometric distortion. The robust similarity measure can effectively eliminate the nonlinear radiation difference caused by multiple sensors. Based on the above reasons, the researchers combined these two types of methods to achieve more accurate matching. Table 3 shows a summary of the improvements on integrated algorithms in MMRS image registration.

**Table 3.** Analysis of integrated method improvements.

| Author | Method Category | Method Improvement | Conclusion |
|---|---|---|---|
| Gong et al. [3] | Combined area-based and feature-based methods | A novel coarse-to-fine scheme for automatic image registration. | In coarse registration (pre-registration), the scale histogram was used to remove the outliers detected by SIFT. Due to the robustness of MI to noise and its adaptability to different image intensity values, adopted it to refine the pre-registration results in the multiresolution framework. |
| Ye et al. [72] | Combined area-based and feature-based methods | A local descriptor-based registration method for multispectral remote sensing images. | Pre-registration used SR-SIFT and projection transformation to eliminate the obvious rotation and scale differences between the reference and sensed images. Then the second stage uses LSS as a new similarity measure. |
| Xiong et al. [91] | Combined area-based and feature-based methods | Coarse registration based on intersections of straight lines and fine registration based on MI from separated patches. | Using the stable and consistent edge information of the optical and SAR images, the registration accuracy is improved. |
| Zhang et al. [92] | Combined area-based and feature-based methods | Using the feature-based SAR-SIFT algorithm to complete the coarse registration, and then utilize the area-based ROEWA-HOG method to complete the fine registration. | Achieve high-precision automatic registration of the hybrid model. |

Gong et al. [3] proposed a novel coarse-to-fine scheme for automatic image registration. In coarse registration (pre-registration), the scale histogram was used to remove the outliers detected by SIFT. The dense clusters in the histogram were the true scale differences between the images. Key point pairs that contribute to the cluster were considered to be correct matches, and those far from the cluster were considered incorrect matches and removed. Due to the robustness of MI to noise and its adaptability to different image intensity values, the authors adopted it to refine the pre-registration results in the multiresolution framework. Finally, the improved Marquardt–Levenberg search strategy was developed to find the global optimum. In [72], the authors put forward a local descriptor-based registration method for multispectral remote sensing images. In the first stage, pre-registration used SR-SIFT and projection transformation to eliminate the obvious rotation and scale differences between the reference and sensed images. In the second stage, first a set of evenly distributed interest points were extracted from the pre-registered image based on the block-Harris method. Then, through the bidirectional matching technique, LSS was used as a new similarity measure for connection point detection (named LSCC). Finally, the pre-registered image was corrected using a piecewise linear model. Experimental results showed that three pairs of multispectral remote sensing image pairs with significant nonlinear intensity differences and geometric distortions from different sensors could achieve reliable registration accuracy.

As mentioned earlier, the edge information is relatively stable and consistent in optical and SAR images, and the line intersection point as a matching primitive is a great choice. Thus, Xiong et al. [91] proposed coarse registration based on intersections of straight lines and fine registration based on MI from separated patches. Zhang et al. [92] used optical and SAR images as the reference image and the sensed image respectively. They first used the feature-based SAR-SIFT algorithm to complete the coarse registration, and then utilized the area-based ROEWA-HOG method to complete the fine registration to achieve high-precision automatic registration of the hybrid model.

### 3.4. Review of Deep Learning-Based Methods

Existing methods of using deep learning to solve the MMRS image registration problem can be roughly divided into three categories. First, deep learning algorithms (such as CNN and GAN) are integrated into the existing MMRS image registration method to generate high-level and deep features. Second, generating an end-to-end network framework based on a Siamese network for MMRS image registration, and third, using deep learning to eliminate the difference between modalities, and then combining traditional methods for image registration. Table 4 summarizes improvements on deep learning-based algorithms for MMRS image registration.

**Table 4.** Analysis of deep learning-based method improvements.

| Author | Method Category | Method Improvement | Conclusion |
|---|---|---|---|
| Ye et al. [93] | Methods to improve CNN and GAN algorithms | Using CNN to extract the middle and high-level features of an image and combined them with the low-level features extracted by SIFT. | Features of CNN and SIFT were incorporated into the PSO-SIFT algorithm for registration. |
| Ma et al. [94] | Methods to improve CNN and GAN algorithms | Using VGG-16 to approximate spatial relationships and proposed a new point matching strategy based on spatial relationships and combined the local feature-based methods. | Due to the powerful feature extraction capabilities of CNN and the consideration of spatial relationships, the matching results are robust and accurate. |
| Yang et al. [95] | Methods to improve CNN and GAN algorithms | Adopting a pre-trained VGG network to generate multi-scale descriptors through high-level convolution information features. | Optimize the registration details by increasing the number of feature points. |
| Quan et al. [96] | Methods to improve CNN and GAN algorithms | Applying GAN for MMRS images data augmentation, which could immensely enhance the accuracy and robust of registration process. | A dual-channel deep network and CNN can save the spatial information of the image. |
| Merkle et al. [97] | Methods to improve CNN and GAN algorithms | A GAN-based method for dealing with optical and SAR image registration. | The generator in GAN accurately and reliably retained the geometric structure of the optical image, opening up new possibilities for MMRS. |
| Hughes et al. [98] | Methods to improve Siamese network | A specific pseudo-Siamese network to register the optical and SAR images. | The two convolutional streams of this network are identical and independent, and there is no parameter sharing to process the different intensity information of the two images. |
| Zhang et al. [99] | Methods to improve Siamese network | A fully convolutional Siamese network, which used an end-to-end training process. It built a general framework for MMRS image registration based on depth features. | Sharing parameters between the two branches to solve the problem of lack of data sets. |

| Author | Method Category | Method Improvement | Conclusion |
|---|---|---|---|
| Merkle et al. [100] | Methods to improve Siamese network | By training the Siamese network to learn the spatial transformation between optical and SAR images. | The effectiveness of this approach for the generation of reliable and robust matching points between optical and SAR images had been demonstrated by experiment. |
| He et al. [101] | Methods to improve Siamese network | A novel multi-scale remote sensing image registration deep network, which includes the following three steps: corner detection based on s-Harris, a search strategy based on Gaussian pyramid coupled quadtree. Finally, global to local quadratic polynomial constraints and RANSAC were utilized to remove mismatches. | This algorithm can realize the scale comparison of multi-scale conjugate patches, make the matching evenly distributed, and obtain satisfactory matching accuracy. The most important thing is to avoid the influence of complex background changes on the registration results. |
| Zhang et al. [102] | Others | Extending the image visual attribute transfer method to pre-process an image. | Eliminate the intensity differences between the multi-modal image pairs, and make the color, texture and other characteristics consistent in a similar structural area. |
| Wang et al. [103] | Others | A remote sensing image registration framework based on deep learning, which directly learns the mapping function between image patch pairs and their matching labels through closed-loop information. | A self-learning method was introduced to avoid a few image data and can learn the mapping function from itself without relying on other data. At the same time, the application of transfer learning further improves the registration accuracy and reduces training costs. |
| Zampieri et al. [104] | Others | Designing a neural network with a specific scale to learn image features. | Complete the alignment between remote sensing images and maps |

### 3.4.1. Methods to Improve CNN and GAN Algorithms

Ye et al. [93] used CNN to extract the middle and high-level features of an image and combined them with the low-level features extracted by SIFT. Since the PSO-SIFT algorithm achieved advanced performance in remote sensing image registration, features of CNN and SIFT were incorporated into the PSO-SIFT algorithm for registration. Visual Geometry Group (VGG) is a commonly used high-level image feature extraction network. Ma et al. [94] used VGG-16 to approximate spatial relationships and proposed a new point matching strategy based on spatial relationships and combined the local feature-based methods. Due to the powerful feature extraction capability of CNN, the traditional matching method is stable and effective. Considering the spatial relationship, the matching result is more robust and accurate. In [95], the authors adopted a pre-trained VGG network to generate multi-scale descriptors through high-level convolution information features and utilized the gradually expanding dynamic inlier selection to optimize the registration details by increasing the number of feature points.

Quan et al. [96] applied GAN for MMRS images data augmentation, which could immensely enhance the accuracy and robust of registration process. The author used a dual-channel deep network as a matching network, which helped to extract different features of optical and SAR images. CNN was introduced as a feature extractor to save the spatial information of the image. Through multiple constraints, such as associative constraints and geometric constraints to delete incorrect matching points. Experimental results showed that this method was superior to traditional methods and had a great registration performance for optical and SAR images. Merkle et al. [97] proposed a GAN-based method for dealing with optical and SAR image registration. First, a pseudo-SAR image was generated from the optical image by training a GAN-based network, so that

the two images had similar intensity information. Subsequently, the area-based method and the feature-based method were introduced to match the pseudo-SAR with the real SAR image. The experimental results validated the necessity of this method, especially the generator in GAN. It accurately and reliably retained the geometric structure of the optical image, opening up new possibilities for MMRS.

### 3.4.2. Methods to Improve Siamese Network

The methods of using deep learning to solve MMRS image registration problems are mostly based on Siamese networks. These methods are usually trained to learn the features and differences between multi-modal image pairs. It is then used to measure the similarity between these image pairs.

Hughes et al. [98] produced a specific pseudo-Siamese network to register the optical and SAR images. The two convolutional streams of this network are identical and independent, and there is no parameter sharing to process the different intensity information of the two images. Finally, the fully connected layer is used to fuse the different features learned by the two branches. While Zhang et al. [99] shared parameters between the two branches to solve the problem of lack of data sets. The author proposed a fully convolutional Siamese network, which used an end-to-end training process. It built a general framework for MMRS image registration based on depth features, which used the image block selected by the Harris detector to input the network, and finally obtained the similarity between the sub-image and the remote sensing image. Merkle et al. [100] trained the Siamese network to learn the spatial transformation between optical and SAR images. First, the features of the two images were extracted through CNN, and then the dot product layer [105] was produced to measure the similarity. The point with the highest final response value was regarded as the matching point. The effectiveness of this approach for the generation of reliable and robust matching points between optical and SAR images had been demonstrated by experiment. He et al. [101] proposed a novel multi-scale remote sensing image registration deep network, which includes the following three steps: corner detection based on s-Harris, a search strategy based on Gaussian pyramid coupled quadtree to narrow the search space and realize multi-scale comparison of conjugate patches. Finally, global to local quadratic polynomial constraints and RANSAC were utilized to remove mismatches. The visualization results showed that this method can make the matching evenly distributed and obtain satisfacting matching accuracy. The most important thing was to avoid the influence of complex background changes on the registration results.

### 3.4.3. Others

In [102], the authors extended the image visual attribute transfer method to pre-process an image in order to eliminate the intensity differences between the multi-modal image pairs, and make the color, texture and other characteristics consistent in a similar structural area. Then, it used traditional local feature-based algorithms for image registration. Wang et al. [103] proposed a remote sensing image registration framework based on deep learning, which directly learns the mapping function between image patch pairs and their matching labels through closed-loop information. A self-learning method was introduced to avoid a few image data and can learn the mapping function from itself without relying on other data. At the same time, the application of transfer learning further improves the registration accuracy and reduces training costs. Zampieri et al. [104] completed the alignment between remote sensing images and maps by designing a neural network with a specific scale to learn image features.

### 3.5. Multi-Modal Image Registration Methods in Other Fields

In other related fields, image processing methods such as photogrammetry, medical imaging, and computer vision have their characteristics and innovations. Some cutting-edge image processing methods will help researchers achieve new breakthroughs. Table 5 summarizes image registration algorithms in other fields.

**Table 5.** Analysis of image registration algorithms in other fields.

| Author | Method Category | Method Improvement | Conclusion |
| --- | --- | --- | --- |
| Tombari et al. [106] | Multimodal registration of photogrammetric images | The local maximum feature of the Contextual Self-dissimilarity (CSD) operator is detected through the non-maximum suppression (NMS) stage, which is called the Maximal Self-Dissimilarity interest point detector (MSD). | MSD is taking the lead for photogrammetric based multimodal registration. |
| Sedghi et al. [107] | Multimodal registration of medical images | Building a five-layer neural network to learn a similarity metric that can measure the level of registration and using Powell's method to optimize the learned metric in an iterative manner. | The results demonstrate the feasibility of learning a useful deep metric from substantially misaligned training data, and results are significantly better than from Mutual Information. |
| Wang et al. [108] | Multimodal registration of medical images | A content-adaptive weakly-supervised deep learning framework is constructed for multi-modal retinal image registration. The framework is composed of three neural networks for blood vessel segmentation, feature detection and description, and outlier elimination. | This method is superior to other learning-based methods, achieved the highest success rate and Dice coefficient, and has significant robustness in poor quality images. |
| Huang et al. [109] | Multimodal registration of medical images | An end-to-end network architecture composed of affine transformation and deformable transformation is proposed. | Double consistency constraints and a new loss function based on prior knowledge have been developed to achieve accurate and efficient multi-contrast MR image registration. |
| Jhan et al. [110] | Multimodal registration of computer vision images | A new N-SURF matching method for multi-spectral camera (MSCs) image registration and a general tool for image registration of various MSCs. | This method has good accuracy and can obtain more the number of correct matches (CMs) that are evenly distributed, which has the advantages of accuracy and efficiency. |
| Zhou et al. [111] | Multimodal registration of computer vision images | A potential generation model for cross-weather image alignment based on intensity constancy and image manifold characteristics. | Experimental results demonstrate that this approach can significantly outperform the state-of-the-art methods. |

Maximal Self-dissimilarity (MSD) [106] is taking the lead for photogrammetric based multimodal registration. The features detected through the Non-Maxima Suppression (NMS) stage are local maxima of the Contextual Self-dissimilarity (CSD) operator, herein the detector will be referred to as Maximal Self-Dissimilarity interest point detector (MSD).

In the field of medical image processing, multi-modal image registration is more mature and advanced. Its main methods can also be divided into similarity measurement problems, which are intensity-based, and control point pair extraction based on features. Using deep learning strategies to solve the above problems has become a new trend. In terms of learning similarity measures, Sedghi et al. [107] built a five-layer neural network to learn a similarity metric that can measure the level of registration to estimate the model parameters in the matching of 3D-US and MR abdominal scans. Then, Powell's method is used to optimize the learned metric in an iterative manner. Wang et al. [108] constructed a content-adaptive weakly supervised deep learning framework for multi-modal retinal image registration, which is composed of three neural networks for blood vessel segmentation, feature detection and description, and outlier elimination. This method is superior to other learning-based methods, achieved the highest success rate and Dice coefficient, and has significant robustness for poor quality images. Huang et al. [109] proposed a novel unsupervised learning-based framework. This is an end-to-end network architecture composed of affine and deformable transformations. In addition, double

consistency constraints and a new loss function based on prior knowledge have been developed to achieve accurate and efficient multi-contrast MR image registration.

For cross-spectral image matching in computer vision, Jhan et al. [110] proposed a new N-SURF matching method for multi-spectral camera (MSCs) image registration. This method has good accuracy and can obtain more correct matches (CMs) that are evenly distributed. Based on the N-SURF and EPT models, the authors developed a general tool for image registration of various MSCs, which can quickly and accurately co-register a large number of MS images, has the advantages of accuracy and efficiency, and does not require prior knowledge on sensor calibration. For cross-temporal image matching, Zhou et al. [111] were inspired by the GAN method and proposed a potential generation model for cross-weather image alignment. Based on intensity constancy and image manifold characteristics, this method describes the image registration task as a constrained deformation flow estimation problem with a potential encoding process.

## 4. Experiments

In this section, we select 8 representative methods and 2 assessment metrics to conduct experiments, which can provide an objective performance reference for different MMRS image registration methods and hence support relative engineering with credible evidence. The experiments are conducted on a computer with 2.5 GHz Intel Core CPU, 16 GB memory, and MATLAB codes. The codes of the 8 representative image registration methods are all publicly available.

We test the 8 representative methods on the 6 MMRS image pairs from CoFSM (CoFSM: https://skyearth.org/publication/project/CoFSM/ (accessed on 10 December 2021)), which consist of modality pairs, namely, depth-optical, optical-optical (cross-temporal), infrared-optical, map-optical, day-night and SAR-optical. The six pairs of original images selected in the database are shown in Figure 5.

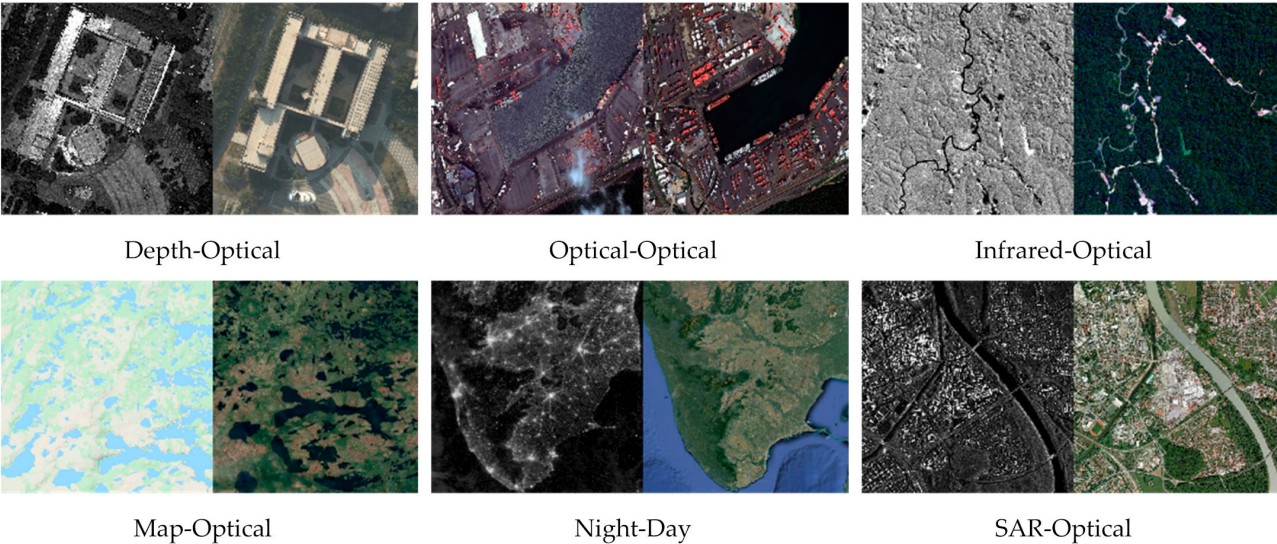

**Figure 5.** Selected original image pairs from the database, which covers six multi-modal pairs in the field of remote sensing.

Figure 6 shows the performances of 8 methods on six pairs of MMRS images. It is easy to find that the traditional single-mode image registration algorithm is inferior to some recently popular improved algorithms in terms of registration performance. SIFT [33,34], SAR-SIFT [112] and PSO-SIFT (SIFT, SAR-SIFT and PSO-SOFT available at https://github.com/ZeLianWen/Image-Registration (accessed on 10 December 2021)) [113] are all based on the image gradient to extract features and construct feature descriptors, which are very sensitive to the strong radiation difference of MMRS images. The core idea of SURF (Available at https://www.mathworks.com/matlabcentral/fileexchange/28300-opensurf-including-image-warp/?ivk_sa=1024320u (accessed on 10 December 2021)) [38]

is similar to SIFT, which is an accelerated version of the SIFT algorithm, and the registration performance is also not good enough. In contrast, HOPC [4], CFOG (Available at https://github.com/yeyuanxin110/CFOG (accessed on 10 December 2021)) [63], RIFT [79] (Available at http://www.escience.cn/people/lijiayuan/index.html (accessed on 10 December 2021)) and RIFT-LAF [114] (Available at https://github.com/StaRainJ/LAF (accessed on 10 December 2021)) methods, which lead the development of MMRS image registration, show better registration performance.

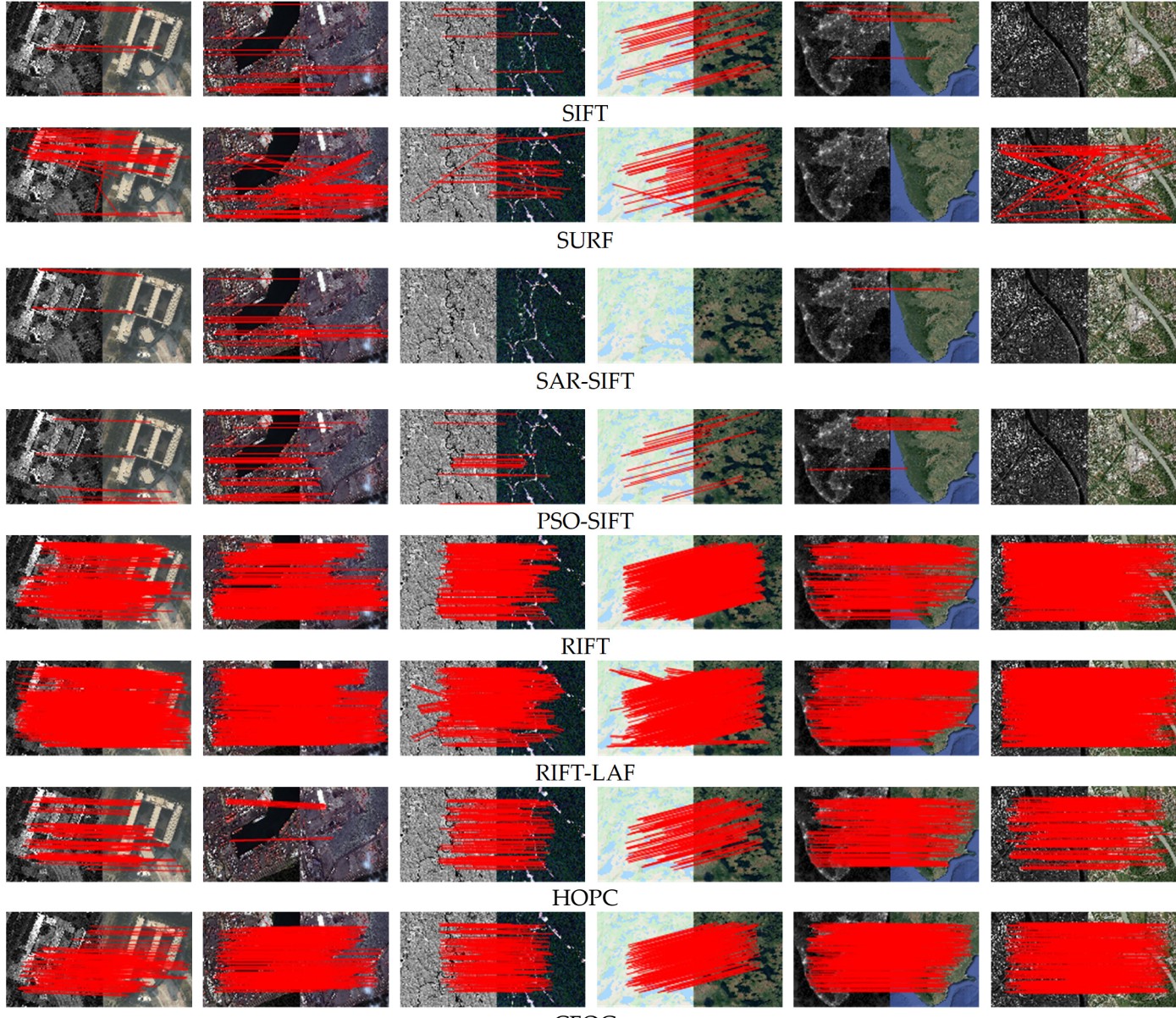

**Figure 6.** The matching results of 8 methods on typical multi-modal image pairs in the remote sensing research field. (No matching result indicates that the method is not applicable to this multi-modal image pair).

In addition, we select two frequently used assessment metrics, namely the correct match ratio (CMR) and root mean square error (RMSE), to evaluate the performances of the different MMRS image registration methods. Tables 6–8 report the results of the two metrics and run times using the 8 representative methods on the 6 image pairs. In each table, we use '/' to indicate that this method cannot be applicable to multimodal image pairs, so there is no data information.

**Table 6.** CMR at different MMRS image pairs.

| Method \ Image Pairs | Depth-Opti. | Opti-Opti. | IR-Opti. | Map-Opti. | Night-Day | SAR-Opti. |
|---|---|---|---|---|---|---|
| SIFT | 7/122 | 15/83 | 6/103 | 35/146 | 6/37 | / |
| SURF | 42/830 | 81/1025 | 23/943 | 65/375 | / | 20/1297 |
| SAR-SIFT | 4/43 | 25/127 | / | / | 4/20 | / |
| PSO-SIFT | 4/137 | 12/113 | 7/64 | 16/124 | 5/49 | / |
| RIFT | 258/1152 | 285/956 | 319/800 | 554/1351 | 238/958 | 596/1520 |
| RIFT-LAF | 607/1152 | 496/956 | 283/800 | 410/1351 | 387/958 | 897/1520 |
| HOPC | 89/219 | 8/186 | 128/230 | 174/498 | 153/202 | 286/328 |
| CFOG | 98/258 | 213/956 | 135/319 | 193/554 | 168/238 | 303/596 |

**Table 7.** RMSE or Precision at different MMRS image pairs.

| Method \ Image Pairs | Depth-Opti. | Opti-Opti. | IR-Opti. | Map-Opti. | Night-Day | SAR-Opti. |
|---|---|---|---|---|---|---|
| SIFT | 0.44 | 0.52 | 0.49 | 0.58 | 0.22 | / |
| SURF | 0.34 | 0.14 | 0.24 | 0.20 | / | 0.21 |
| SAR-SIFT | 0.12 | 0.62 | / | / | 0.06 | / |
| PSO-SIFT | 0.35 | 0.64 | 0.55 | 0.56 | 0.63 | / |
| RIFT | 0.64 | 0.63 | 0.64 | 0.70 | 0.68 | 0.66 |
| RIFT-LAF | 0.30 | 0.36 | 0.53 | 0.59 | 0.38 | 0.40 |
| HOPC | / | / | / | / | / | / |
| CFOG | / | / | / | / | / | / |

**Table 8.** Operation times at different MMRS image pairs.

| Method \ Image Pairs | Depth-Opti. | Opti-Opti. | IR-Opti. | Map-Opti. | Night-Day | SAR-Opti. |
|---|---|---|---|---|---|---|
| SIFT | 13.69 | 14.04 | 8.39 | 12.90 | 11.44 | / |
| SURF | 2.69 | 2.24 | 2.15 | 1.69 | / | 2.55 |
| SAR-SIFT | 15.02 | 12.59 | / | / | 11.66 | / |
| PSO-SIFT | 12.91 | 12.86 | 9.17 | 10.28 | 9.34 | / |
| RIFT | 10.93 | 9,31 | 8.32 | 13.04 | 9.63 | 11.03 |
| RIFT-LAF | 11.04 | 9.41 | 8.65 | 13.25 | 9.87 | 11.31 |
| HOPC | 61.65 | 61.50 | 40.24 | 55.60 | 62.69 | 62.99 |
| CFOG | 10.93 | 4.29 | 1.30 | 3.95 | 2.73 | 2.47 |

From the results, we can see that the performance of classic descriptors (SIFT, SURF, etc.) is not perfect for most multimodal image pairs, and the number of CM of these methods is significantly less than the improved algorithm based on multimodality. Especially for SAR-optical image pairs, due to the significant non-linear intensity difference between the two modalities and the presence of SAR image speckle noise, both have a great impact on the registration results of the classic method. In our experiments, both RIFT and RIFT-based LAF matching algorithms have shown satisfactory performance. However, it should be noted that the RIFT algorithm used in the experiment does not consider rotation invariance. First, it needs to be unified with the geographic registration technology used by other methods. Second, when the rotation invariance is considered in the model, the computation time is very long. The HOPC algorithm based on structural similarity achieves good performance on multi-modal image pairs, but it performs poorly on cross-temporal

(optical-optical) image pairs. As a multi-modal image template matching framework, CFOG has excellent performance in matching performance and computational efficiency.

## 5. Challenges and Future Directions

Through the review of the most cutting-edge MMRS image registration method in the previous section, current research has largely solved the nonlinear radiation distortion caused by modal differences and the geometric distortions caused by various factors. However, there are still several issues that need further consideration:

1.  There is insufficient data for different modal conditions in the field of remote sensing, and there is no complete and comprehensive database containing all types of MMRS image pairs. Therefore, for deep learning methods that are developing rapidly and showing great progress, most of the work uses this technology for feature detection and feature description. The lack of training and test data greatly limits its application in MMRS image registration.

2.  With the development of remote sensing technology, images from different sensors will have higher resolution. The details within high-resolution images are complex, the amount of data is large, and the local geometric deformation caused by the undulation of the terrain cannot be ignored. This results in a significant challenge for image registration.

3.  By comparing existing methods on MMRS image registration, we found that there are many registration strategies for optical-to-SAR and infrared-to-visible. However, learning-based methods can only learn the characteristics and differences between specific multi-modal image pairs. In the future, how to fuse and learn the data between multi-modalities should become the focus. For example, rather than eliminating the modal difference between two images, the registration method should be suitable for remote sensing images with multiple modalities.

4.  Area-based methods still face the impact of overlapping block regions and low computational efficiency on the registration performance, whereas feature-based methods face the challenge of nonlinear intensity differences.

In future research, equipping the MMRS image database and improving the registration performance through deep learning will become more mainstream. Of course, the improvement and fusion of traditional methods can also further improve the registration accuracy, and better integrate image registration into other imaging applications. Furthermore, future research trends will focus on such as solving the problem of multi-modal image pair registration at the same time through data fusion and so on. As for multi-modal high-resolution remote sensing images, how to improve the registration accuracy under significant geometric deformation and global radiation difference will be a tough challenge.

## 6. Conclusions

In recent years, MMRS image registration has attracted widespread attention given its important imaging applications, such as image fusion and target recognition [115]. Thus, this article comprehensively investigates existing MMRS image registration methods, which can be divided into three categories: area-based, feature-based and deep learning-based. Through a brief introduction to the major methods and theoretical frameworks, this paper provides a basis for further research to improve the performance of MMRS image registration. The latest research results are classified according to image registration methods, providing future research ideas for people in related fields. In addition, we use 8 representative methods to perform experimental evaluation on 6 common multi-modal remote sensing image pairs, and visually demonstrate the performance of different methods for MMRS image registration.

**Author Contributions:** Conceptualization, X.Z. and Y.H.; methodology, X.Z. and C.L.; formal analysis, X.Z., C.L., A.B. and Z.P.; investigation, X.Z. and Y.H.; resources, X.Z. and C.L.; data curation, C.L., Z.P.; writing—original draft preparation, X.Z. and Y.H.; writing—review and editing, C.L., A.B. and I.C.; visualization, X.Z. and Y.H.; supervision, C.L., I.C. and Z.P.; funding acquisition, C.L., A.B., I.C.; All authors have read and agreed to the published version of the manuscript.

**Funding:** This work is supported by the National Natural Science Foundation of China under Grant Nos. 61702251, 61971273, the Key Research and Development Program in Shaanxi Province under Grant no. 2021GY-032, in part by the Natural Sciences and Engineering Research Council of Canada, and Youth Academic Talent Support Program of Northwest University under Grant No. 360051900151.

**Institutional Review Board Statement:** Not applicable.

**Informed Consent Statement:** Not applicable.

**Data Availability Statement:** Publicly available datasets were analyzed in this study. This data can be found here: https://skyearth.org/publication/project/CoFSM/ (accessed on 10 December 2021).

**Conflicts of Interest:** The authors declare no conflict of interest.

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
