# Peer review of "Multimodal Remote Sensing Image Registration Methods and Advancements: A Survey"

_remotesensing, doi:10.3390/rs13245128_

Round 1

Reviewer 1 Report

The work must be well presented, although addressing the following few clarifications would help.

1- Compared this survey with other surveys in this topic.  

The main contribution of this paper is not well written, and author(S)  have not provided adequate motivation and it clearly does not make a reasonable original contribution.

2- Presentation style of paper has many weakness, so it confuse the reader:

  • The title of the section 2 and 3 (e.g. Area-based methods , Area-based methods improvements)!

Author Response

Comments to the Author

The work must be well presented, although addressing the following few clarifications would help.

  1. Compared this survey with other surveys in this topic.  

The main contribution of this paper is not well written, and author(S) have not provided adequate motivation and it clearly does not make a reasonable original contribution.

Response:

Thanks very much for your comments. As a review article, the innovation contribution of this article is indeed not outstanding enough, and the novelty and importance of this review is not clearly stated in the introduction of the manuscript. In view of this, we have revised the part of introduction in RED to emphasize the motivation of this article. The motivation of this article is to fill up the gap of multi-modal images in remote sensing image registration. In terms of summarizing cutting-edge methods, the popular learning-based methods are introduced, and the methods are listed in an orderly manner, so that readers are clear at a glance.

Revised part:

Existing surveys on remote sensing image registration mostly include single-modal general registration frameworks and matching methods, while MMRS image registration as a very important branch, only occupies a small part of the article [1,6-8]. A large part of the multi-modal image processing work is focused on medical image registration [9-11], and few works specifically review the part of multi-modal images in remote sensing image processing. In this regard, we review the general methods of MMRS image registration, especially classification according to the registration method category, and introduce recently popular learning-based methods, so that readers can learn about cutting-edge methods in the field at a glance.

  1. Presentation style of paper has many weakness, so it confuse the reader:

The title of the section 2 and 3 (e.g. Area-based methods, Area-based methods improvements)!

Response:

Thanks for your helpful comments. The section 2 of this article mainly introduces the general framework of MMRS image registration method, which is divided into area-based method, feature-based method and deep learning-based method. The subsections in each method are intended to illustrate each step within the framework of the method. The section 3 focuses on reviewing the different improvement algorithms under the framework of each method to facilitate readers to understand the differences and connections between the improved methods.

We have tried our best to revise the writing to improve readability. we have revised the title of sections in RED in the manuscript, and correspondingly, we have modified the integrate structural framework in Figure 1:

  1. General framework of MMRS image registration method

2.1. General framework of area-based methods

2.2. General framework of feature-based methods

2.3. General scheme of deep learning-based methods

  1. Review of MMRS image registration methods

3.1. Review of area-based method

3.1.1. Methods to improve similarity measures

3.1.2. Methods to improve optimization

3.1.3. Hybrid methods

3.2. Review of Feature-based method

3.2.1. Methods to improve SIFT or Harris operator

3.2.2. Methods to improve feature descriptor

3.2.3. Hybrid methods

3.3. Review of combined area-based and feature-based methods

3.4. Review of deep learning-based methods

3.4.1. Methods to improve CNN and GAN algorithms

3.4.2. Methods to improve Siamese network

3.4.3. Others

Figure 1. Structural framework of this review.

In addition to the above issues, we have tried our best to modify other parts of the article, such as:

  1. We have expanded image registration methods in other fields (including medical imaging, computer vision, and photogrammetry) to make this survey more complete.
  2. We have conducted 8 representative comparison experiments on 6 pairs of MMRS images in section 4, and listed commonly used data sets and open source codes, which make the article more intuitively present the advantages and disadvantages of different algorithms.
  3. The challenges that still exist in MMRS image registration are emphasized to make future research directions more specific and contextual.

Reviewer 2 Report

General comments:

This submission is a proposal for a review paper on the topic of multimodal image registration in the specific field of remote sensing, and more specifically for satellite or long range imaging.

The contribution is solid and complete and the methodology developed to browse methodology and improvements structure is clear and efficient. All references given seems relevant, even if it reveals a strong scope focuses, on domain (remote-sensing), technique (satellite) and imaging (SAR to something of reverse). Some other possibles improvement or clarification are given and detailed below. The conclusion is a bit  sharp and factual and miss a perspective ( except for the expected explosion of DL method that could be mention in most of current research papers…) and/or critical distance. For example the following sentence/statement, « However, there is a lack of systematic and general framework applicable to all multimodal types » could or should be better discussed, positioned or contextualized. One way could be for example to link or expend this image-registration review to data fusion issues and challenges (several articles in the literature).

Detailed comments  :

A ) Scope/domain limit issue

Inter or cross domain analysis is a hard but always valuable task for review papers. The most effective, useful and cited review articles are the ones that breaks disciplinary, domain or technical frontiers. On this point, this proposal deepen a strongly limited context (satellite), and only a narrow technical range, not all the component of remote-sensing is touched by this article. For this reason a limitation section or paragraph must be added at the end of the introduction.  As en example, close-range applications of remote-sensing (photogrammetry etc) face similar issues, they are not included but at the same time they cannot directly benefit or learn from this generous study.

B ) Possible complement

In some other related fields, astrology, medical imaging, computer vision or cultural heritage image pre-processing is widely used to improve image registration, it could be basic image/signal processing method or more advanced filtering (Wallis filter, etc). The paper will gain in completeness if a dedicated sub-section is added (like the one on other DL method not used for remote-sensing). As en example of out of boundaries review studies, Maximal Self-dissimilarity (MSD) is taking the lead for photogrammetric based multimodal registration.

Minor comments :

- Correct some misunderstandings, errors or complement (see annotated PDF)

Motivation of the decision :

I suggest a minor-revision to correct/complement most if not all the point above-mentioned.

Author Response

Comments to the Author

This submission is a proposal for a review paper on the topic of multimodal image registration in the specific field of remote sensing, and more specifically for satellite or long range imaging.

The contribution is solid and complete and the methodology developed to browse methodology and improvements structure is clear and efficient. All references given seems relevant, even if it reveals a strong scope focuses, on domain (remote-sensing), technique (satellite) and imaging (SAR to something of reverse). Some other possibles improvement or clarification are given and detailed below.

  1. The conclusion is a bit  sharp and factual and miss a perspective ( except for the expected explosion of DL method that could be mention in most of current research papers) and/or critical distance. For example the following sentence/statement, « However, there is a lack of systematic and general framework applicable to all multimodal types » could or should be better discussed, positioned or contextualized. One way could be for example to link or expend this image-registration review to data fusion issues and challenges (several articles in the literature).

Response:

Thanks very much for your helpful comments. “However, there is a lack of systematic and general framework applicable to all multimodal types” is a bit sharp and general. We have started from the positioning of data fusion and discussed the challenges still faced in the field of MMRS image registration. The revised conclusion is highlighted in GREEN:

However, learning-based methods can only learn the characteristics and differences between specific multi-modal image pairs. In the future, how to fuse and learn the data between multi-modalities should become the focus. For example, rather than eliminating the modal difference between two images, the registration method should be suitable for remote sensing images with multiple modalities.

  1. Detailed comments: A) Scope/domain limit issue

Inter or cross domain analysis is a hard but always valuable task for review papers. The most effective, useful and cited review articles are the ones that breaks disciplinary, domain or technical frontiers. On this point, this proposal deepen a strongly limited context (satellite), and only a narrow technical range, not all the component of remote-sensing is touched by this article. For this reason a limitation section or paragraph must be added at the end of the introduction.  As en example, close-range applications of remote-sensing (photogrammetry etc) face similar issues, they are not included but at the same time they cannot directly benefit or learn from this generous study.

Response:

Thanks for your constructive comments. We have supplemented the introduction based on your suggestion, which is highlighted in GREEN:

The field of remote sensing can be roughly divided into remote sensing image acquisition technology and remote sensing information processing technology. The photogrammetry technology that mainly relies on aerospace and ground imaging platforms is different from the remote sensing technology that relies on satellite platforms. There are also differences in imaging bands and imaging methods. Photogrammetry is mainly to obtain accurate geographic information from remote sensing, and its application also has the above-mentioned problems. Therefore, remote sensing image acquisition technology and close-range application of photogrammetry cannot benefit from this survey.

  1. Detailed comments: B) Possible complement

In some other related fields, astrology, medical imaging, computer vision or cultural heritage image pre-processing is widely used to improve image registration, it could be basic image/signal processing method or more advanced filtering (Wallis filter, etc). The paper will gain in completeness if a dedicated sub-section is added (like the one on other DL method not used for remote-sensing). As en example of out of boundaries review studies, Maximal Self-dissimilarity (MSD) is taking the lead for photogrammetric based multimodal registration.

Response:

Thanks for your comments. Section 2.3.3 in the manuscript briefly introduced other DL methods not used for remote sensing.

As a supplement, we have tried our best to collect and present image processing methods in other related fields before submitting the manuscript. However, because of the cross-professional and cross-field, we apologize for the lack of comprehensive coverage, and follow-up research will definitely make up for this deficiency. The added section is in section 3.5, which is highlighted in GREEN in the manuscript.

3.5. Multi-modal image registration methods in other fields

In other related fields, image processing methods such as photogrammetry, medical imaging, and computer vision have their characteristics and innovations. Some cutting-edge image processing methods will help researchers achieve new breakthroughs. Table 5 summarizes image registration algorithms in other fields.

Table 5. Analysis of image registration algorithms in other fields.

Author

Method category

Method improvement

Conclusion

Tombari et al. [106]

Multimodal registration of photogrammetric images

The local maximum feature of the Contextual Self-dissimilarity (CSD) operator is detected through the non-maximum suppression (NMS) stage, which is called the Maximal Self-Dissimilarity interest point detector (MSD).

MSD is taking the lead for photogrammetric based multimodal registration.

Sedghi et al. [107]

Multimodal registration of medical images

Building a five-layer neural network to learn a similarity metric that can measure the level of registration, and using Powell’s method to optimize the learned metric in an iterative manner.

The results demonstrate the feasibility of learning a useful deep metric from substantially misaligned training data, and results are significantly better than from Mutual Information.

Wang et al. [108]

Multimodal registration of medical images

A content-adaptive weakly-supervised deep learning framework is constructed for multi-modal retinal image registration. The framework is composed of three neural networks for blood vessel segmentation, feature detection and description, and outlier elimination.

This method is superior to other learning-based methods, achieved the highest success rate and Dice coefficient, and has significant robustness in poor quality images.

Huang et al. [109]

Multimodal registration of medical images

An end-to-end network architecture composed of affine transformation and deformable transformation is proposed.

Double consistency constraints and a new loss function based on prior knowledge have been developed to achieve accurate and efficient multi-contrast MR image registration.

Jhan et al. [110]

Multimodal registration of computer vision images

A new N-SURF matching method for multi-spectral camera (MSCs) image registration and a general tool for image registration of various MSCs.

This method has good accuracy and can obtain more the number of correct matches (CMs) that are evenly distributed, which has the advantages of accuracy and efficiency.

Zhou et al. [111]

Multimodal registration of computer vision images

A potential generation model for cross-weather image alignment based on intensity constancy and image manifold characteristics.

Experimental results demonstrate that this approach can significantly outperform the state-of-the-art methods.

Maximal Self-dissimilarity (MSD) [106] is taking the lead for photogrammetric based multimodal registration. The features detected through the Non-Maxima Suppression (NMS) stage are local maxima of the Contextual Self-dissimilarity (CSD) operator, herein the detector will be referred to as Maximal Self-Dissimilarity interest point detector (MSD).

In the field of medical image processing, multi-modal image registration is more mature and advanced. Its main methods can also be divided into similarity measurement problems, which are intensity-based, and control point pair extraction based on features. Using deep learning strategies to solve the above problems has become a new trend. In terms of learning similarity measures, Sedghi et al. [107] built a five-layer neural network to learn a similarity metric that can measure the level of registration to estimate the model parameters in the matching of 3D-US and MR abdominal scans. Then, Powell's method is used to optimize the learned metric in an iterative manner. Wang et al. [108] constructed a content-adaptive weakly supervised deep learning framework for multi-modal retinal image registration, which is composed of three neural networks for blood vessel segmentation, feature detection and description, and outlier elimination. This method is superior to other learning-based methods, achieved the highest success rate and Dice coefficient, and has significant robustness for poor quality images. Huang et al. [109] proposed a novel unsupervised learning-based framework. This is an end-to-end network architecture composed of affine and deformable transformations. In addition, double consistency constraints and a new loss function based on prior knowledge have been developed to achieve accurate and efficient multi-contrast MR image registration.

For cross-spectral image matching in computer vision, Jhan et al. [110] proposed a new N-SURF matching method for multi-spectral camera (MSCs) image registration. This method has good accuracy and can obtain more correct matches (CMs) that are evenly distributed. Based on the N-SURF and EPT models, the authors developed a general tool for image registration of various MSCs, which can quickly and accurately co-register a large number of MS images, has the advantages of accuracy and efficiency, and does not require prior knowledge on sensor calibration. For cross-temporal image matching, Zhou et al. [111] were inspired by the GAN method and proposed a potential generation model for cross-weather image alignment. Based on intensity constancy and image manifold characteristics, this method describes the image registration task as a constrained deformation flow estimation problem with a potential encoding process.

  1. Minor comments:

Correct some misunderstandings, errors or complement (see annotated PDF).

Response:

Thanks very much for your careful revision in the manuscript, we have revised one by one according to your comments in GREEN in the manuscript.

In order to emphasize the difference between CNN and GAN in the deep learning framework module, the following sentences have been added in GREEN:

Unlike CNN's powerful ability to analyze data and extract features, GANs focus on generating data, enhancing data by the adversarial network, or generating fake images to eliminate modal differences.

Reviewer 3 Report

In this review paper the authors investigate the existing multi-modal remote sensing image registration methods. The topic is interesting and matches well for MDPI Remote Sensing journal. The paper contains meaningful review of related works. However the paper has some unclear points, and the following minor concerns.

1. In my opinion, the paper would benefit significantly if the authors would separate information about the features of each of the pairwise two modal image registrations in a separate section. For example, what difficulties arise when registering optical and LIDAR images, and so on - e.g. optical-to-SAR, infrared-to-visual, and visual-to-map…?

2. There are also typos in the paper.

- 3.1.1, 3.1.2, 3.1.3 have the same title

- L. 195 - «are very popular technologists»

- L. 297 - «log-Gabor filter» and L. 299 «Log-Gabor filters»

Author Response

Comments to the Author

In this review paper the authors investigate the existing multi-modal remote sensing image registration methods. The topic is interesting and matches well for MDPI Remote Sensing journal. The paper contains meaningful review of related works. However the paper has some unclear points, and the following minor concerns.

  1. In my opinion, the paper would benefit significantly if the authors would separate information about the features of each of the pairwise two modal image registrations in a separate section. For example, what difficulties arise when registering optical and LIDAR images, and so on - e.g. optical-to-SAR, infrared-to-visual, and visual-to-map…

Response:

Thank you for your helpful comments and affirmation of our work. Firstly, the theme of this paper is the review of MMRS image registration methods and improved methods. Therefore, the framework of this paper is based on the classification of registration methods. For example, the section 3 is a review of feature-based methods, which is divided into the research of improved SIFT and Harris operator, the research of improved feature descriptor and so on. Your opinion is to classify and discuss separately according to the MMRS image pairs, what are their representative methods and the difficult challenges that still need to be solved. This is a great idea. We will consider exploring from this perspective in future research.

Second, in order to improve our article according to your opinions, we have made a supplement in the Introduction, mainly introducing several common modal image pair registration methods, emphasizing the focus of the framework of this paper, and providing literature guidance for researchers who want to know more about some multimodal registration methods, which emphasizes in ORANGE in the manuscript:

Common MMRS images include cross-temporal, cross-season, optical to SAR, optical to infrared, optical to LIDAR, map to visible, etc. In the existing literature, research on optical-SAR [12-14] and optical-infrared [15-17] is most common. We discuss registration methods for MMRS images, rather than the types of modal image pairs. Readers who want to know about more registration methods can refer to [18].

Third, we have added experiments of 6 different model image pairs in Section 4, and compared eight representative registration methods. Readers can also more intuitively understand the registration performance and difficulties/challenges between images of different modalities from the experimental data.

  1. There are also typos in the paper.

- 3.1.1, 3.1.2, 3.1.3 have the same title

- L. 195 - «are very popular technologists»

- L. 297 - «log-Gabor filter» and L. 299 «Log-Gabor filters»

Response:

Thanks for your helpful comments. we have revised these errors in RED and ORANGE in the manuscript.

- We are sorry that we did not find the same title in Section 3.1, 3.2 and 3.3 due to our carelessness. We have revised the title in the manuscript to: 3.1.1. Methods to improve similarity measures, 3.1.2. Methods to improve optimization and 3.1.3. Hybrid methods.

- We have changed «are very popular technologists» to «is a very popular technology».

- We have unified L. 297-«log-Gabor filter» and L. 299 «Log-Gabor filters» as Log-Gabor filters, because here it means multiple Log-Gabor filters with different center frequencies.

Reviewer 4 Report

Multi-modal remote sensing image registration is very important recently. The manuscript presents comprehensive reviews and analysis of MMRS image registration methods. Some issues are:
1. Review of current image registration survey articles.
2. I suggest add some experiments of different methods.
3. The challenges should be highlighted.
4. The commonly used datasets should be listed.
5. The open source code could be added.
6. The general registration scheme should be illustrated.
7. More latest articles should be added, i.e., the year of 2021.

Author Response

Comments to the Author

Multi-modal remote sensing image registration is very important recently. The manuscript presents comprehensive reviews and analysis of MMRS image registration methods. Some issues are:

  1. Review of current image registration survey articles.

Response:

Thanks for your helpful comments. we have revised the part of introduction in RED to review existing image registration surveys:

[6] J. Ma, X. Jiang, A. Fan, J. Jiang, J. Yan, Image matching from handcrafted to deep features: A survey, Int. J. Comput. Vis. 1–57, 2020.

[7] S. Dawn, V. Saxena, B. Sharma, Remote sensing image registration techniques: A survey, in: Proceedings of the International Conference on Image and Signal Processing, pp. 103–112, 2010.

[8] Y. Wu, J.W. Liu, C.Z. Zhu, Z.F. Bai, Q.G. Miao, W.P. Ma and M.G. Gong. Computational Intelligence in Remote Sensing Image Registration: A survey. International Journal of Automation and Computing, 18(1), 2020.

[9] G. Haskins, U. Kruger, P. Yan, Deep learning in medical image registration: a survey, Mach. Vis. Appl. 31(1), 2020.

[10] A. Sotiras, C. Davatzikos, N. Paragios, Deformable medical image registration: A survey, IEEE Trans. Med. Imaging, vol. 32, no. 7, pp. 1153–1190, 2013.

[11] Y. Fu, Y. Lei, T. Wang, W.J. Curran, T. Liu, X. Yang, Deep learning in medical image registration: a review, Phys. Med. Biol. 2020.

And then, we have illustrated the motivation of this article, which is to fill up the gap of multi-modal images in remote sensing image registration. In terms of summarizing cutting-edge methods, the popular learning-based methods are introduced, and the methods are listed in an orderly manner, so that readers are clear at a glance.

  1. I suggest add some experiments of different methods.

Response:

Thanks for your constructive comments. We have added some comparative experiments in section 4, which is highlighted in BLUE:

  1. Experiments

In this section, we select 8 representative methods and 2 assessment metrics to conduct experiments, which can provide an objective performance reference for different MMRS image registration methods and hence support relative engineering with credible evidence. The experiments are conducted on a computer with 2.5 GHz Intel Core CPU, 16 GB memory, and MATLAB codes. The codes of the 8 representative image registration methods are all publicly available.

We test the 8 representative methods on the 6 MMRS image pairs from CoFSM1, which consist of modality pairs, namely, depth-optical, optical-optical (cross-temporal), infrared-optical, map-optical, day-night and SAR-optical. The six pairs of original im-ages selected in the database are shown in Figure 5.

Figure 5. Select the original image pairs from the collected database, which covers six multi-modal pairs in the field of remote sensing research.

Figure 6 shows the performances of 8 methods on six pairs of MMRS images. It is easy to find that the traditional single-mode image registration algorithm is inferior to some recently popular improved algorithms in terms of registration performance. SIFT [33,34], SAR-SIFT [112] and PSO-SIFT [113] are all based on the image gradient to extract features and construct feature descriptors, which are very sensitive to the strong radiation difference of MMRS images. The core idea of SURF [38] is similar to SIFT, which is an accelerated version of the SIFT algorithm, and the registration performance is also not good enough. In contrast, HOPC [4], CFOG [67], RIFT [87] and RIFT-LAT [114] methods, which lead the development of MMRS image registration, show better registration performance.

Figure 6. The matching results of 8 methods on typical multi-modal image pairs in the remote sensing research field. (No matching result indicates that the method is not applicable to this multi-modal image pair).

In addition, we select two frequently used assessment metrics, i.e., the correct match ration (CMR) and root mean square error (RMSE), to evaluate the performances of the different MMRS image registration methods. Table 6,7 and Table 8 report the results of the two metrics and times using the 8 representative methods on the 6 image pairs. In each table, we use ‘/’ to indicate that this method cannot applicable to multi-modal image pairs, so there is no data information.

Table 6. CMR at different MMRS image pairs.

Image pairs

Method

Depth-Opti.

Opti-Opti.

IR-Opti.

Map-Opti.

Night-Day

SAR-Opti.

SIFT

7/122

15/83

6/103

35/146

6/37

/

SURF

42/830

81/1025

23/943

65/375

/

20/1297

SAR-SIFT

4/43

25/127

/

/

4/20

/

PSO-SIFT

4/137

12/113

7/64

16/124

5/49

/

RIFT

258/1152

285/956

319/800

554/1351

238/958

596/1520

RIFT-LAF

607/1152

496/956

283/800

410/1351

387/958

897/1520

HOPC

89/219

8/186

128/230

174/498

153/202

286/328

CFOG

98/258

213/956

135/319

193/554

168/238

303/596

Table 7. RMSE or Precision at different MMRS image pairs.

Image pairs

Method

Depth-Opti.

Opti-Opti.

IR-Opti.

Map-Opti.

Night-Day

SAR-Opti.

SIFT

0.44

0.52

0.49

0.58

0.22

/

SURF

0.34

0.14

0.24

0.20

/

0.21

SAR-SIFT

0.12

0.62

/

/

0.06

/

PSO-SIFT

0.35

0.64

0.55

0.56

0.63

/

RIFT

0.64

0.63

0.64

0.70

0.68

0.66

RIFT-LAF

0.30

0.36

0.53

0.59

0.38

0.40

HOPC

/

/

/

/

/

/

CFOG

/

/

/

/

/

/

Table 8. Operation times at different MMRS image pairs.

Image pairs

Method

Depth-Opti.

Opti-Opti.

IR-Opti.

Map-Opti.

Night-Day

SAR-Opti.

SIFT

13.69

14.04

8.39

12.90

11.44

/

SURF

2.69

2.24

2.15

1.69

/

2.55

SAR-SIFT

15.02

12.59

/

/

11.66

/

PSO-SIFT

12.91

12.86

9.17

10.28

9.34

/

RIFT

10.93

9,31

8.32

13.04

9.63

11.03

RIFT-LAF

11.04

9.41

8.65

13.25

9.87

11.31

HOPC

61.65

61.50

40.24

55.60

62.69

62.99

CFOG

10.93

4.29

1.30

3.95

2.73

2.47

From the results, we can see that the performance of classic descriptors (SIFT, SURF, etc.) is not perfect for most multimodal image pairs, and the number of CM of these methods is significantly less than the improved algorithm based on multimodality. Especially for SAR-optical image pairs, due to the significant non-linear intensity difference between the two modalities and the presence of SAR image speckle noise, both have a great impact on the registration results of the classic method. In our experiments, both RIFT and RIFT-based LAF matching algorithms have shown satisfactory performance. However, it should be noted that the RIFT algorithm used in the experiment does not consider rotation invariance. First, it needs to be unified with the geographic registration technology used by other methods. Second, when the rotation invariance is considered in the model, the computation time is very long. The HOPC algorithm based on structural similarity achieves good performance on multi-modal image pairs, but it performs poorly on cross-temporal (optical-optical) image pairs. As a multi-modal image template matching framework, CFOG has excellent performance in matching performance and computational efficiency.

  1. The challenges should be highlighted.

Response:

Thanks for your comments. We have discussed and specified the challenges of MMRS image registration, which is highlighted in BLUE and GREEN:

For deep learning technology, discuss its future trends and challenges from the perspective of data lack and data fusion:

  1. There is insufficient data for different modal conditions in the field of remote sensing, and there is no complete and comprehensive database containing all types of MMRS image pairs. Therefore, for deep learning methods that are developing rapidly and showing great progress, most of the work uses this technology for feature detection and feature description. The lack of training and test data greatly limits its application in MMRS image registration.
  2. By comparing existing methods on MMRS image registration, we found that there are many registration strategies for optical-to-SAR and infrared-to-visible. However, learning-based methods can only learn the characteristics and differences between specific multi-modal image pairs. In the future, how to fuse and learn the data between multi-modalities should become the focus. For example, rather than eliminating the modal difference between two images, the registration method should be suitable for remote sensing images with multiple modalities.

Combining the characteristics of high-resolution remote sensing images, analyze the current situation and future trends and challenges of the problem:

  1. With the development of remote sensing technology, images from different sensors will have higher resolution. The details within high-resolution images are complex, the amount of data is large, and the local geometric deformation caused by the undulation of the terrain cannot be ignored. This results in a significant challenge for image registration.

Finally, summarize and highlight the challenges mentioned:

In future research, equipping the MMRS image database and improving the registration performance through deep learning will become more mainstream. Of course, the improvement and fusion of traditional methods can also further improve the registration accuracy, and better integrate image registration into other imaging applications. Furthermore, future research trends will focus on such as solving the problem of multi-modal image pair registration at the same time through data fusion and so on. As for multi-modal high-resolution remote sensing images, how to improve the registration accuracy under significant geometric deformation and global radiation difference will be a tough challenge.

  1. The commonly used datasets should be listed.

Response:

Thanks for your helpful comments. We used six pairs of MMRS images in the added comparative experiment in Section 4, all of which are from the CoFSM dataset, which have been listed in the footnotes of the manuscript.

CoFSM: https://skyearth.org/publication/project/CoFSM/.

  1. The open source code could be added.

Response:

Thanks for your helpful comments. We have used 8 representative remote sensing image registration methods in the added comparative experiments. The codes of the 8 methods are all publicly available, which have been listed in the footnotes in the manuscript.

SIFT, SAR-SIFT and PSO-SOFT available at https://github.com/ZeLianWen/Image-Registration

SURF available at https://www.mathworks.com/matlabcentral/fileexchange/28300-opensurf-including-image-warp/?ivk_sa=1024320u

HOPC available at https://github.com/yeyuanxin110/HOPC

CFOG available at https://github.com/yeyuanxin110/CFOG

RIFT available at http://www.escience.cn/people/lijiayuan/index.html

LAF available at https://github.com/StaRainJ/LAF

  1. The general registration scheme should be illustrated.

Response:

Thanks for your comments. Firstly, the 2.1 and 2.2 sections of the manuscript introduced the general framework of area-based and feature-based image registration methods, respectively. The framework of area-based method consists of three major contents, including similarity measures, geometric transformation models and optimizations. Feature-based MMRS image registration includes main three steps, which are feature detection, feature description and feature matching. Figure 1. and Figure 2. respectively show the general framework of the image registration method with a flowchart.

Secondly, we apologize for the unclear title that has bothered readers. We have changed the titles of Section 2.1 and Section 2.2 to 2.1. General framework of area-based methods, 2.2. General framework of feature-based methods.

  1. More latest articles should be added, i.e., the year of 2021.

Response:

Thanks for your constructive comments. We have added seven latest 2021 articles, which are highlighted in the references:

[18] Jiang, X.; Ma, J.; Xiao, G.; Shao, Z.; Guo. X. A review of multimodal image matching: methods and applications. Information Fusion. 2021, 11.

[68] Liang, D.; Ding, J.; Zhang, Y. Efficient Multisource Remote Sensing Image Matching Using Dominant Orientation of Gradient. IEEE Journal of Selected Topics in Applied Earth Observations and Remote Sensing. 2021, 14, 2194-2205.

[90] Xu, G.; Wu, Q.; Cheng, Y.; Yan, F.; Li Z.; Yu. Q. A robust deformed image matching method for multi-source image matching. Infrared Physics & Technology, 2021, 115, 103691.

[108] Wang, Y.; Zhang, J. K.; Cavichini, M.; Bartsch, D. G.; Freeman, W. R.; Nguyen, T. Q.; An, C. Robust Content-Adaptive Global Registration for Multimodal Retinal Images Using Weakly Supervised Deep-Learning Framework. IEEE Transactions on Image Processing, 2021, 30, 3167–3178.

[109] W. Huang, H. Yang, X. Liu, C. Li, I. Zhang, R. Wang, H. Zheng and S. Wang. A Coarse-to-Fine Deformable Transformation Framework for Unsupervised Multi-Contrast MR Image Registration with Dual Consistency Constraint. IEEE Transactions on Medical Imaging, vol. 40, no. 10, pp. 2589-2599, 2021.

[110] Jhan, J. P.; Rau. J. Y. A Generalized Tool for Accurate and Efficient Image Registration of UAV Multi-lens Multispectral Cameras by N-SURF Matching. IEEE Journal of Selected Topics in Applied Earth Observations and Remote Sensing, 2021, 14, 6353-6362.

[115] Li, Y.; Ma, J.; Zhang, Y. Image retrieval from remote sensing big data: A survey, Inf. Fusion 2021, 67, 94–115.

Round 2

Reviewer 1 Report

 English language and minor spell check required

Author Response

Comments to the Author

English language and minor spell check required.

Response:

Thanks very much for your helpful comments. We have tried our best to check the English language and spell, which are revised in the manuscript in RED, such as:

-  Line 16 – We have changed «analysis» to «analyses»;

-  Line 20 – We have changed «are» to «is»;

-  Line 35 – We have changed «have» to «has»……

Reviewer 4 Report

Most of my previous questions have been well addressed. Challenges and opportunities should be pointed out in independent sections, but not mixed together in the conclusion. A careful grammar check should be conducted.

Author Response

Comments to the Author

Most of my previous questions have been well addressed. Challenges and opportunities should be pointed out in independent sections, but not mixed together in the conclusion. A careful grammar check should be conducted.

Response:

Thanks very much for your helpful comments. We have separated “challenges and future directions” and “conclusion” as Section 5 and Section 6, respectively. Correspondingly, we have modified the text frame of Figure 1, which is highlighted in GREEN:

  1. Challenges and future directions

Through the review of the most cutting-edge MMRS image registration method in the previous section, current research has largely solved the nonlinear radiation distortion caused by modal differences and the geometric distortions caused by various factors. However, there are still several issues that need further consideration:

  1. There is insufficient data for different modal conditions in the field of remote sensing, and there is no complete and comprehensive database containing all types of MMRS image pairs. Therefore, for deep learning methods that are developing rapidly and showing great progress, most of the work uses this technology for feature detection and feature description. The lack of training and test data greatly limits its application in MMRS image registration.
  2. With the development of remote sensing technology, images from different sensors will have higher resolution. The details within high-resolution images are complex, the amount of data is large, and the local geometric deformation caused by the undulation of the terrain cannot be ignored. This results in a significant challenge for image registration.
  3. By comparing existing methods on MMRS image registration, we found that there are many registration strategies for optical-to-SAR and infrared-to-visible. How-ever, learning-based methods can only learn the characteristics and differences between specific multi-modal image pairs. In the future, how to fuse and learn the data between multi-modalities should become the focus. For example, rather than eliminating the modal difference between two images, the registration method should be suitable for remote sensing images with multiple modalities.
  4. Area-based methods still face the impact of overlapping block regions and low computational efficiency on the registration performance; whereas feature-based methods face the challenge of nonlinear intensity differences.

In future research, equipping the MMRS image database and improving the registration performance through deep learning will become more mainstream. Of course, the improvement and fusion of traditional methods can also further improve the registration accuracy, and better integrate image registration into other imaging applications. Furthermore, future research trends will focus on such as solving the problem of multi-modal image pair registration at the same time through data fusion and so on. As for multi-modal high-resolution remote sensing images, how to improve the registration accuracy under significant geometric deformation and global radiation difference will be a tough challenge.

  1. Conclusion

In recent years, MMRS image registration has attracted widespread attention given its important imaging applications, such as image fusion and target recognition [115]. Thus, this article comprehensively investigates existing MMRS image registration methods, which can be divided into three categories: area-based, feature-based and deep learning-based. Through a brief introduction to the major methods and theoretical frameworks, this paper provides a basis for further research to improve the performance of MMRS image registration. The latest research results are classified according to image registration methods, providing future research ideas for people in related fields. In addition, we use 8 representative methods to perform experimental evaluation on 6 common multi-modal remote sensing image pairs, and visually demonstrate the performance of different methods for MMRS image registration.

Figure 1. Structural framework of this review.

In addition, we have tried our best to check the grammar, which are revised in the manuscript in RED, such as:

-  Line 16 – We have changed «analysis» to «analyses»;

-  Line 20 – We have changed «are» to «is»;

-  Line 35 – We have changed «have» to «has»……